# Impacts of the global food system on terrestrial biodiversity from land use and climate change

Elizabeth H. Boakes [1,2] ✉, Carole Dalin [2,3], Adrienne Etard [1,4] & Tim Newbold [1]

The global food system is a key driver of land-use and climate change which in turn drive biodiversity change. Developing sustainable food systems is therefore critical to reversing biodiversity loss. We use the multi-regional input-output model EXIOBASE to estimate the biodiversity impacts embedded within the global food system in 2011. Using models that capture regional variation in the sensitivity of biodiversity both to land use and climate change, we calculate the land-driven and greenhouse gas-driven footprints of food using two metrics of biodiversity: local species richness and rarity-weighted species richness. We show that the footprint of land area underestimates biodiversity impact in more species-rich regions and that our metric of rarity-weighted richness places a greater emphasis on biodiversity costs in Central and South America. We find that methane emissions are responsible for 70% of the overall greenhouse gas-driven biodiversity footprint and that, in several regions, emissions from a single year's food production are associated with global biodiversity loss equivalent to 2% or more of that region's total land-driven biodiversity loss. The measures we present are relatively simple to calculate and could be incorporated into decision-making and environmental impact assessments by governments and businesses.

Anthropogenic pressures continue to drive biodiversity loss despite increasing conservation efforts[1]. Land use is currently the greatest driver of biodiversity change[1,2], and has thus been the focus of much previous conservation research. However, the impacts of climate change on biodiversity are expected to increase considerably[3], and, by 2070, climate may match or surpass land use as the greatest driver of biodiversity change[4,5]. Agricultural land used for food production covers an estimated 38–55% of Earth's habitable land[6,7], while the global food system is responsible for 21–37% of anthropogenic greenhouse gas (GHG) emissions[8–10] and could add nearly 1 °C to warming by 2100[11]. The development of sustainable food systems will be critical in halting/reversing land-driven[12] and climate-driven biodiversity loss[11,13].

Increasingly, food grown in one country is traded internationally to satisfy demand elsewhere[14–16]. The associated biodiversity impact of this international consumption is driven by upstream economic activities that are often geographically distant from the locations of biodiversity loss[17]. Reducing the environmental impacts of food therefore requires both supply-side and demand-side changes[17].

Understanding the separate contributions of agricultural land use and greenhouse gas emissions to biodiversity loss is an important step toward developing a sustainable international trade in food. A strategy which considers land use alone might well differ from one that incorporates GHG-driven biodiversity loss. For example, the land and GHG footprint per tonne of crops arising from $N_2O$ tend to be negatively

[1]Centre for Biodiversity and Environment Research, Department of Genetics, Evolution and Environment, University College London, Gower Street, London, UK. [2]Institute for Sustainable Resources, Bartlett School of Environment, Energy and Resources, University College London, Central House, 14 Upper Woburn Place, London, UK. [3]Laboratoire de Géologie de L'École Normale Supérieure, PSL Research University, UMR8538 CNRS, Paris, France. [4]International Institute for Applied Systems Analysis, Schlossplatz 1, A-2361, Laxenburg, Austria. ✉e-mail: e.boakes@ucl.ac.uk

correlated – intensive farming uses less land than extensive farming but has higher GHG emissions ($N_2O$) arising from higher fertiliser use[18]. Moreover, land use and climate differ in their spatial impacts on biodiversity. Land use affects biodiversity that is local to the land conversion. In contrast, climate change affects biodiversity globally, no matter the location of the emissions source.

Environmentally extended multi-regional input-output models (EEMRIOs) are used to link downstream environmental impacts to upstream drivers, allowing the 'footprints' of commodities to be followed back through complex supply chains[19]. The footprint of a hamburger, for example, would contain impacts not only from cattle but also from cattle feed, fertiliser, machinery, water, packaging, fuel etc.[20]. Summing the impacts associated with each upstream product would, in theory, give the hamburger's total footprint. However, in reality, this simple addition process is impossible given the complications of tracking supply chains, double counting of recycled products, and infinite loops, for example, electricity production requiring water, which requires electricity to pump it. EEMRIOs resolve these problems by using input-output tables to infer production recipes, allocating environmental costs to sectors to avoid double counting and approximating infinite sums with the Leontief inverse matrix[20]. Regional production data and their associated environmental costs are combined with information on international trade, allowing the calculation of a variety of consumption footprints, e.g., land use[21], GHG emissions[22] and water[23].

EEMRIO models have been used to follow the effects of goods and services along global supply chains to estimate biodiversity footprints. Lenzen et al.[24] performed the first global biodiversity footprint analysis, using the IUCN Red List to count species' threats within trade regions. This method assumes that species are equally threatened across their ranges, and excludes non-threatened species. Other methods use a similar philosophy to calculate footprints based on biodiversity threat hotspots[25], bird ranges and the number of individual birds lost from an area[17]. An alternative method, derived from land-use data, uses the countryside species-area relationship (cSAR) to estimate the potential number of extinctions caused by land conversion and international trade[16,26,27]. However, since many of the extinctions have yet to be realised, it is unclear how these extinctions would be allocated to different drivers across time, both past and future[17,28]. Wilting et al.[29] take a step further, deriving biodiversity footprints from GHG emissions as well as land use, using the biodiversity metric Mean Species Abundance (MSA) (although climate impacts are based on model predictions of changes in species richness, so land use and climate impacts are not entirely comparable). The GLOBIO 3.5 biodiversity model[30], on which Wilting et al.[29] analysis was based, assumes that the effects of land use and climate change are even across all terrestrial regions; in reality, biodiversity tends to be more sensitive to land use and climate change in tropical regions[31,32]. Marquadt et al.[28] show that biodiversity footprints based on local (e.g., MSA) versus regional (cSAR) measures of biodiversity differ considerably.

We build on these prior analyses, introducing three additional aspects. (i) We calculate the biodiversity impacts of agricultural land use and GHG-emission footprints using models that directly output metrics of terrestrial biodiversity change in the same units, allowing the drivers' impacts to be compared and splitting emissions into carbon dioxide ($CO_2$), methane ($CH_4$) and nitrous oxide ($N_2O$). (ii) We consider a change in local rarity-weighted species richness relative to an unimpacted baseline in addition to local species richness. Species richness, although easy to measure, captures only one of the many dimensions of biodiversity, and does not always decline with global biodiversity loss[33]. Rarity-weighted richness gives greater weight to species with small geographic range size (range size correlates with species extinction risks[34]) and so declines if rare species are replaced by more common ones. (iii) We use biodiversity models that allow us to capture regional variation in the sensitivity of biodiversity both to

land-use differences and to climate change[31]. We base our biodiversity metrics on local measures of biodiversity averaged across the relevant agricultural areas as opposed to a value averaged across an entire exporting region, meaning that we better account for the wide variation in species richness that occurs within regions. Nevertheless, there will still likely be substantial variation in biodiversity responses within our agricultural aggregations. We calculate the biodiversity change associated with all of the land area used in food production in 2011 and assume that the biodiversity change associated with land conversion is immediate. GHG emissions that are released during land conversion are not considered, because, without detailed land-history knowledge, we cannot estimate the proportion of emissions that have dissipated since conversion, nor apportion food-production emissions across years. To put this gap in our coverage of GHGs into context, direct emissions from agriculture contribute 5.1–6.1 Pg $CO_2$-eq,/yr while the clearing of native land for agriculture contributes around 5.9 (SD 2.9) Pg $CO_2$-eq/yr[35]. Consequently, our ratio of land-driven to GHG-driven biodiversity change compares the impacts of the centuries-long process of global agricultural land conversion to the impacts associated with just a single year of GHG emissions.

We examine the international production-based and consumption-based footprints of food-related commodities in 2011 in terms of: a) land area; b) species richness (land-driven and GHG-driven); and c) rarity-weighted species richness (land-driven and GHG-driven). Production-based footprints are based on the total impacts associated with the products produced within a region, whereas consumption-based footprints are the total impacts associated with the products consumed within that region. We calculate footprints for 33 food-related products that span food's journey from field to farm gate to household to waste in 49 regions (44 countries and 5 rest of world regions). We ask whether the estimated impacts we calculate using the different footprint types (i.e., land-driven versus GHG-driven; and land area versus species richness versus rarity-weighted richness) would lead to the same broad policy recommendations with respect to sustainable production, consumption and trade. We identify the regions and food groups with the highest biodiversity footprints, examining the contributions of land use and GHG emissions to these footprints. We also explore biodiversity footprints per $km^2$ (production) and per capita (consumption) for each region and look at the proportion of regions' consumption footprints that are imported. Our analysis provides a detailed examination into the pathways by which regions' consumption of particular food-related products affects biodiversity worldwide, giving insight into the trade-offs between land use and GHG emissions, and into priorities for demand-side changes.

## Results
### Regional production and consumption-based footprints for total food
For all footprints, regions with a high production-based footprint tend to have a high consumption-based footprint. Our land-driven biodiversity footprints show a different picture from a simple land-area footprint. Important information is missed if the area of agricultural land alone is used as a proxy for the impact of food production on biodiversity (Fig. 1). Differences between land-associated footprints are driven in part by differences in species richness between regions' agricultural areas but also by our characterisation factors, which consider biome-specific sensitivities to land use. Furthermore, footprints also vary according to the measure of biodiversity used. The regions with the greatest land-area footprints in 2011 were Rest of World (RoW) Africa, China, and RoW Asia & Pacific (Fig. 1a). However, while RoW Africa also has the highest land-driven species richness footprint, Brazil and RoW Central & South America (RoW CS America) have the second and third highest species richness footprints, respectively (Fig. 1b). Indeed, RoW CS America hosts many narrow-ranged species, which is reflected in its high rarity-weighted richness footprint (Fig. 1c). Decisions regarding

land use and food production should therefore consider not only the land area involved but also the land-driven loss of different facets of biodiversity (see Supplementary Fig. 5 and Supplementary Data 7 for the footprints of individual products within regions).

India and China are the regions responsible for the highest GHG-driven biodiversity loss, followed by RoW Africa (Fig. 1). However, compared to India and China, RoW Africa has a much lower GHG-driven footprint relative to its land-driven biodiversity footprint. We expected the GHG-driven footprint to be much lower than the land-driven footprint since it relates to the global biodiversity loss caused by a single year of emissions whereas our land-driven footprint relates to the historic conversion of all agricultural land used in 2011. The ratio of land-driven to GHG-driven biodiversity loss varied by region from 16 for rarity-weighted richness production footprint in Russia to 855 for production in RoW C&S America, with several regions, including China, India and RoW Asia, having ratios around 50. Finding ratios of 50 or lower is concerning as it shows that direct emissions from a single year of a region's food production will cause biodiversity loss equivalent to 2% or more of the biodiversity loss caused by that

region's total historic land use. Furthermore, we substantially underestimate biodiversity losses from GHG emissions since our analysis does not include emissions from land clearance.

Some regions are net importers of land-driven biodiversity loss but net exporters of GHG-driven biodiversity loss, e.g., India, or vice versa, e.g., Indonesia. RoW Asia & Pacific, RoW CS America, Australia and Mexico are all net exporters of both land-driven and GHG-driven biodiversity meaning their international exports are harming both their domestic biodiversity and, via climate change, global biodiversity. China, the United States, Russia and RoW Middle East are net importers of both footprints (Fig. 1). The top ten footprints stem from regions with very large land areas and/or populations and, aside from Russia, do not include regions in continental Europe.

## Production-based footprints for world regions aggregated by food-related sector
Production-based footprints vary considerably among food-related groups and, within food-related groups, among aggregated world regions (Fig. 2). As for total food footprints, land-driven biodiversity

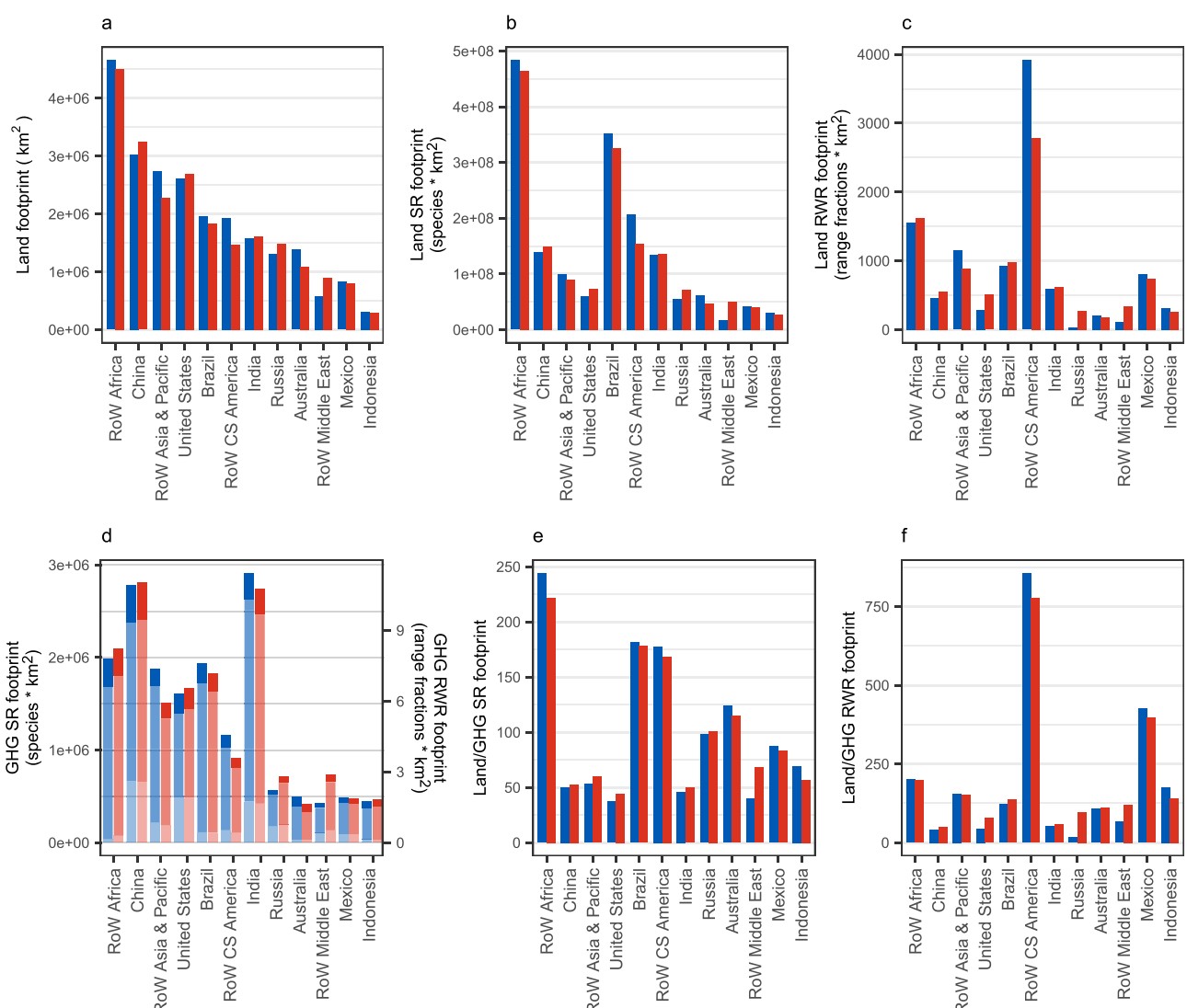

**Fig. 1 | Total production-based (blue bars) and consumption-based (red bars) footprints of food-related products for the year 2011.** Footprints relate to (**a**) land area, (**b**) land-driven species richness (SR) loss, (**c**) land-driven rarity-weighted richness (RWR) loss, (**d**) GHG-driven biodiversity loss split by emissions type: carbon dioxide (dark blue/red), methane (mid blue/red), nitrous oxide (light blue/red) (right-hand axis – species richness; left-hand axis – rarity-weighted richness), (**e**) the ratio of land-driven species richness loss to total GHG-driven species richness loss and (**f**) the ratio of land-driven rarity-weighted richness loss to total GHG-driven rarity-weighted richness loss. Regions which are in the highest ten for one or more footprints are shown. RoW Rest of World. Source data are provided as a source data file.

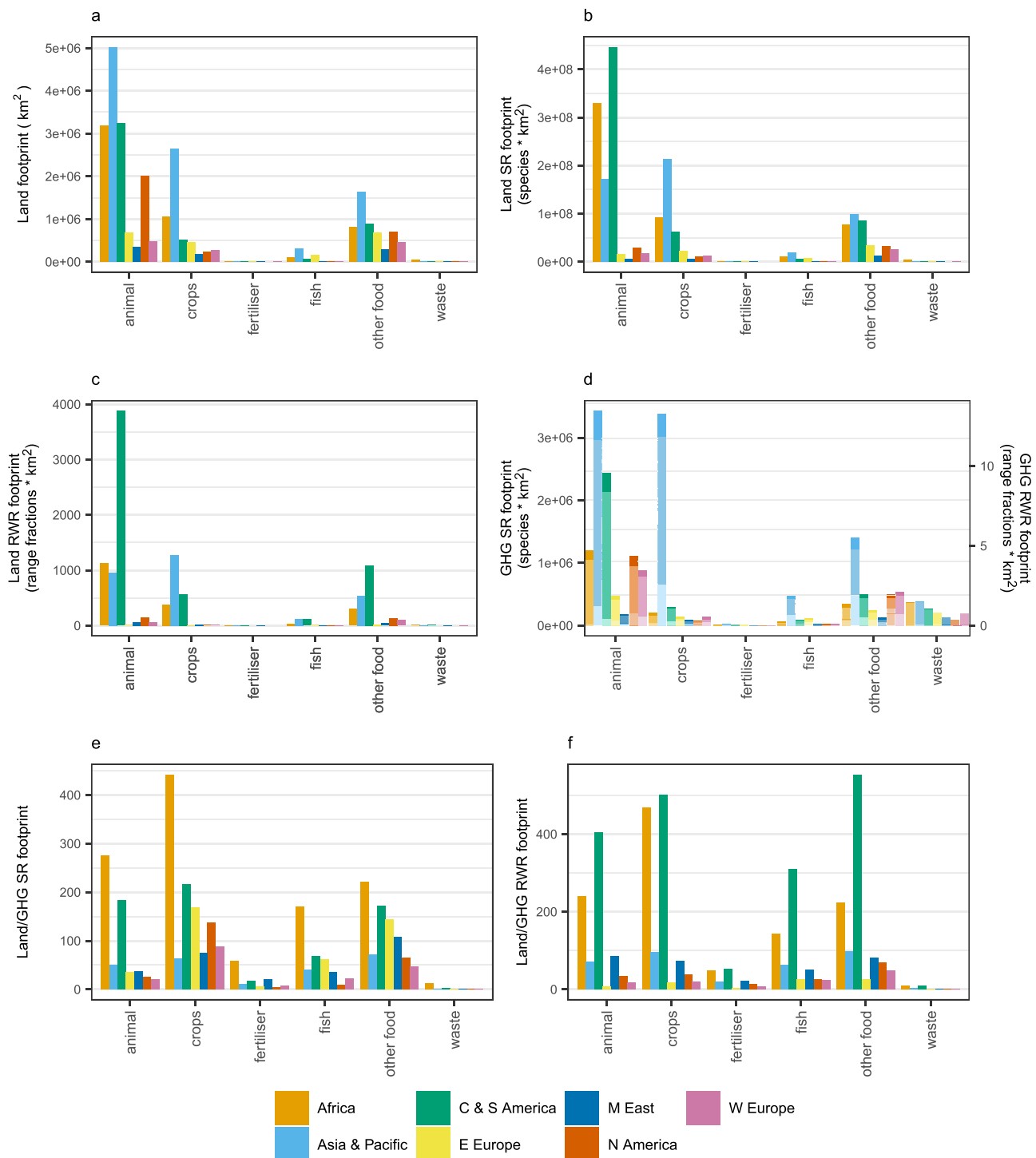

**Fig. 2 | Production-based footprints of aggregated food-related groups within aggregated world regions for the year 2011.** Footprints relate to (**a**) land area, (**b**) land-driven species richness (SR) loss, (**c**) land-driven rarity-weighted richness (RWR) loss, (**d**) GHG-driven biodiversity loss split by emissions type: carbon dioxide (dark shade), methane (mid shade), nitrous oxide (light shade) (right-hand axis – species richness; left-hand axis – rarity-weighted richness), (**e**) the ratio of land-driven species richness loss to total GHG-driven species richness loss and (**f**) the ratio of land-driven rarity-weighted richness loss to total GHG-driven rarity-weighted richness loss. Source data are provided as a source data file.

footprints do not mirror land-area footprints, and the relative size of biodiversity footprints is affected by the richness metric. Using animal-derived products as an example, Asia & Pacific's land footprint is 60% higher than CS America's (and Africa's, Fig. 2a), but its land-driven species-richness footprint is less than half (Fig. 2b), and its rarity-weighted richness footprint less than a third of that of CS America (Fig. 2c). This result reflects Asia & Pacific's lower average natural

species richness compared to CS America's and the lower biodiversity sensitivity of some of its biomes to agricultural land use.

Again, GHG-driven and land-driven biodiversity footprints show different patterns, with tropical regions no longer having consistently higher footprints. Production in Western Europe and North America also drives relatively high GHG-driven biodiversity loss, particularly in the 'Other Food' sector although Asia & Pacific has the highest GHG-

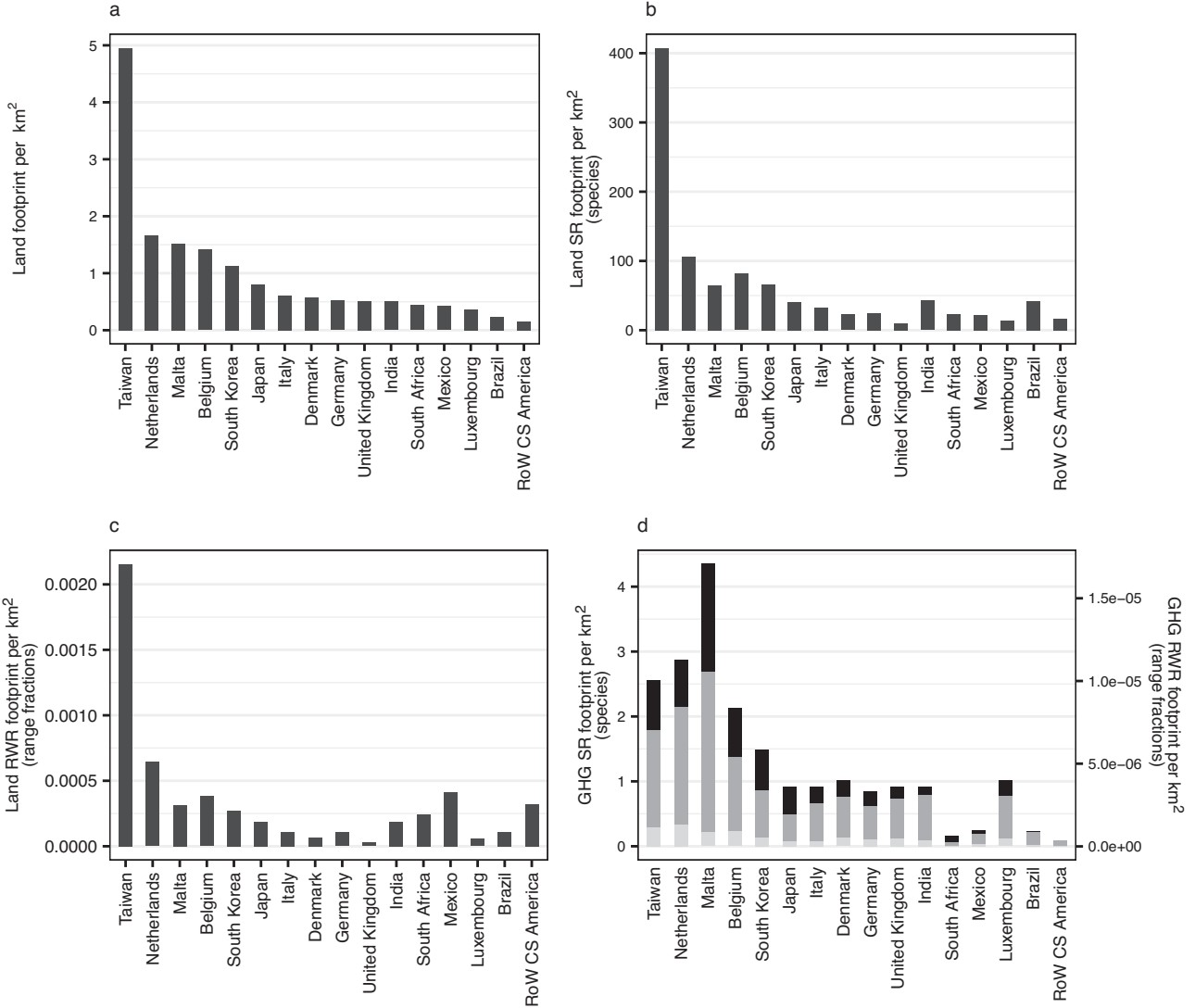

**Fig. 3 | Total per-area production-based footprints of food-related products for the year 2011.** Footprints relate to (**a**) land use, (**b**) land-driven species richness (SR) loss, (**c**) land-driven rarity-weighted richness (RWR) loss and (**d**) GHG-driven biodiversity loss split by emissions type: carbon dioxide (black), methane (dark grey), nitrous oxide (light grey) (right-hand axis - species richness; left-hand axis - rarity-weighted richness). Regions that are in the highest ten for one or more footprints are shown. RoW Rest of World. Source data are provided as a source data file.

driven footprints across all food groups (Fig. 2d). Asia & Pacific's ratio of land-driven to GHG-driven ratios ranges from around 40–100 (Fig. 2e, f) which is extremely concerning given the high land-driven biodiversity footprints this region has. If emissions continue at this rate their impact on global biodiversity will, in under a century, equal that already caused by land conversion in Asia & Pacific.

Animal-derived products tend to have much higher footprints than plant-derived (i.e., crops), although Asia & Pacific is a notable exception for both land-driven and GHG-driven biodiversity footprints, in part due to the production of paddy rice, wheat and other cereals in India (for individual product footprints see Supplementary Fig. 5 and Supplementary Data 8). As would be expected, land-driven biodiversity loss from fertiliser production and the processing of food waste is much lower than that stemming from the production of food itself since the processes use very little agricultural land.

In many instances, the 'Other Food' category has higher production-based GHG-driven biodiversity footprints than the footprints from crops (Fig. 2a–d). Other Food includes 'Food products not elsewhere classified (nec)' (coded in EXIOBASE as 'OFOD') and beverages ('BEVR'). 'Food products nec' contributes the vast majority of

the Other Food footprint and covers a broad range of processed foods such as soups, sandwiches and sauces. The EXIOBASE land-use types that contribute most to the Food Products nec land-driven footprints are cereal grains nec, cattle pasture, wheat, oil seeds, raw milk pasture and fruit/vegetables/nuts (Supplementary Fig. 6).

## Per-area and per-capita footprints

Taiwan stands out as having the highest per-area land-related production-based footprints (Fig. 3a–c). The agriculturally dense but biodiversity-poor UK is an example of a region with relatively lower per-area land-driven biodiversity footprints than per-area land-use footprints (Fig. 3). In contrast, the species-rich Brazil has a lower per-area land-use footprint but a relatively higher per-area land-driven species richness footprint. Some regions (e.g., Brazil, India, Mexico) have both high total production-based footprints (Fig. 1) and high per-area production-based footprints (Fig. 3). The other regions with high per-area production tend to be smaller regions with wide agricultural coverage, for example the Netherlands and Belgium. Small agricultural regions have the highest per-area production-based GHG-driven footprints, for example Taiwan, the Netherlands, and Malta (although see ref. 36 regarding possible over-estimation of Taiwan's

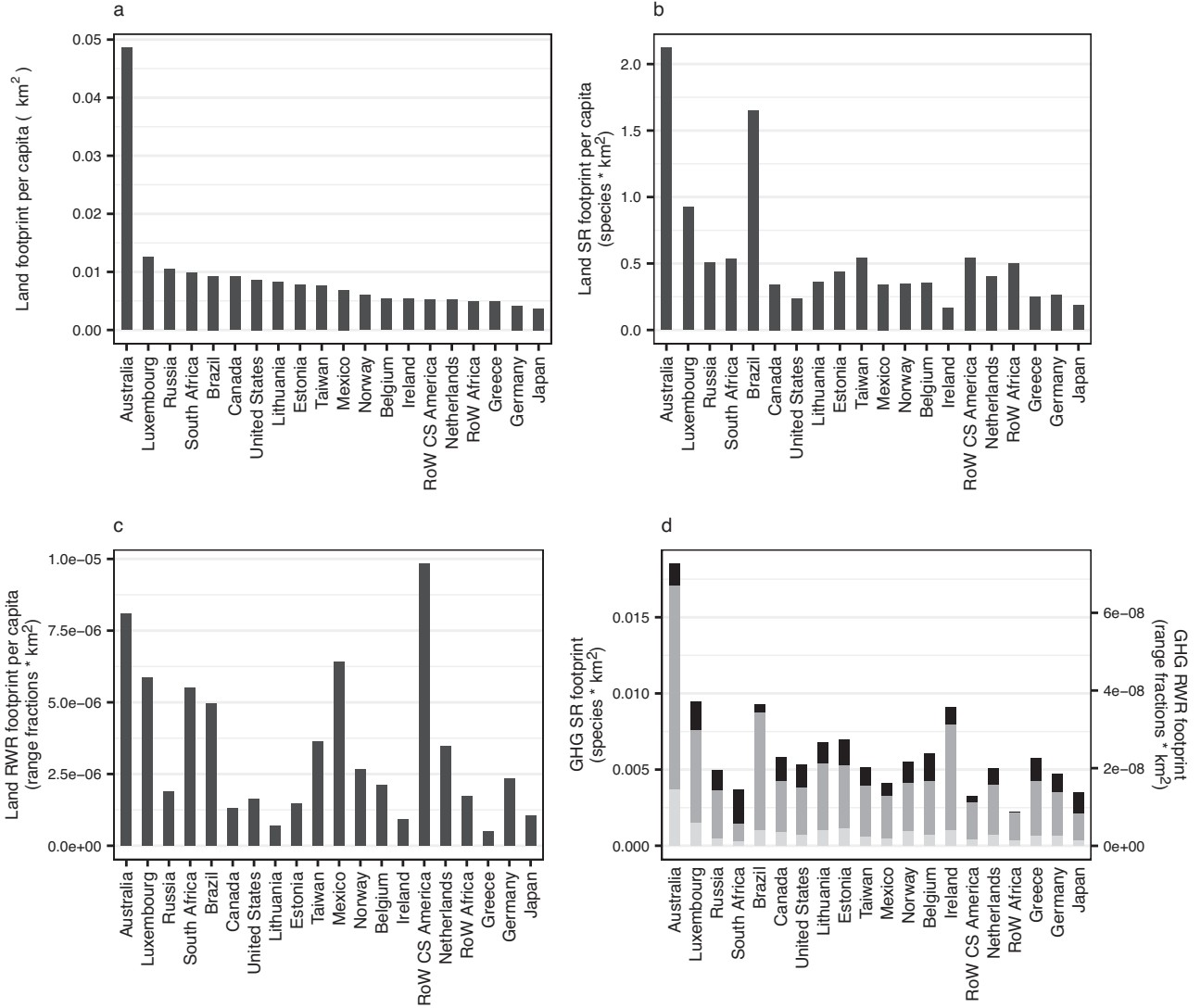

**Fig. 4 | Total consumption-based footprints per-capita of food-related products for the year 2011.** Footprints are in terms of (**a**) land use, (**b**) land-driven species richness (SR) loss, (**c**) land-driven rarity-weighted richness (RWR) loss and (**d**) GHG-driven biodiversity loss split by emissions type: carbon dioxide (black), methane (dark grey), nitrous oxide (light grey) (right-hand axis – species richness; left-hand axis – rarity-weighted richness). Regions that are in the highest ten for one or more footprints are shown. RoW Rest of World. Source data are provided as a source data file.

material footprint). For per-area footprints for all products and regions see Supplementary Fig. 7 and Supplementary Data 9.

Australia is notable for having extremely high land-driven consumption-based footprints per capita (Fig. 4), driven particularly by animal-based products (notably 'Products of Cattle'), but also by plant-based products and 'Other Food' (Supplementary Fig. 8, Supplementary Table 6, and Supplementary Data 10). Luxembourg also has consistently high per-capita consumption-based footprints, probably in part due to its highly affluent population[37]. In contrast to Australia, Luxembourg's consumption-based footprint is primarily driven by plant-derived products (fruit/vegetables/nuts, oil seeds and other crops - i.e., coffee, cocoa, spices). Despite having high total consumption due to their large populations (Fig. 1), China, India and RoW Asia & Pacific are not within the top ten highest per-capita consumption-based footprints for any category. However, some regions with high total biodiversity consumption also have high per-capita footprints: Australia, Brazil, Mexico, Russia, the United States, RoW Africa and RoW CS America. Again, we see contrasts between different footprint types. For example, Brazil's land-driven species richness consumption-based footprint per capita is much larger than RoW CS America's, but the latter's per-

capita rarity-weighted footprint is almost twice that of Brazil (Fig. 4b, c). (Brazil has particularly high per-capita species richness footprints relative to RoW CS America for cattle, processed cattle, vegetable oils and dairy, see Supplementary Fig. 8). Regions with relatively low species richness can have high per-capita land-driven biodiversity consumption-based footprints as a result of importing food.

The products underpinning high per-capita consumption-based GHG-driven biodiversity footprints vary among regions (Fig. 4d). Belgium's high footprint, for example, is largely driven by other foods (e.g., pizza, ready-meals, sauces) and dairy products, Estonia's by dairy products, raw milk, cattle and other foods, Luxembourg's by fruit/vegetable/nuts, other crops (i.e., coffee, cocoa, spices), cattle and chemical fertilizer, and the Republic of Ireland's by processed cattle products and food waste disposed of in landfill. Some regions, e.g., Australia, Luxembourg and Lithuania, have high footprints across multiple products, and at least one animal-based product is associated with particularly high consumption footprints for all regions in Fig. 4, except Taiwan (<-0.0004 species x km² per capita for every animal-based product) (see also Supplementary Fig. 8, Supplementary Table 6 and Supplementary Data 10 for further breakdowns).

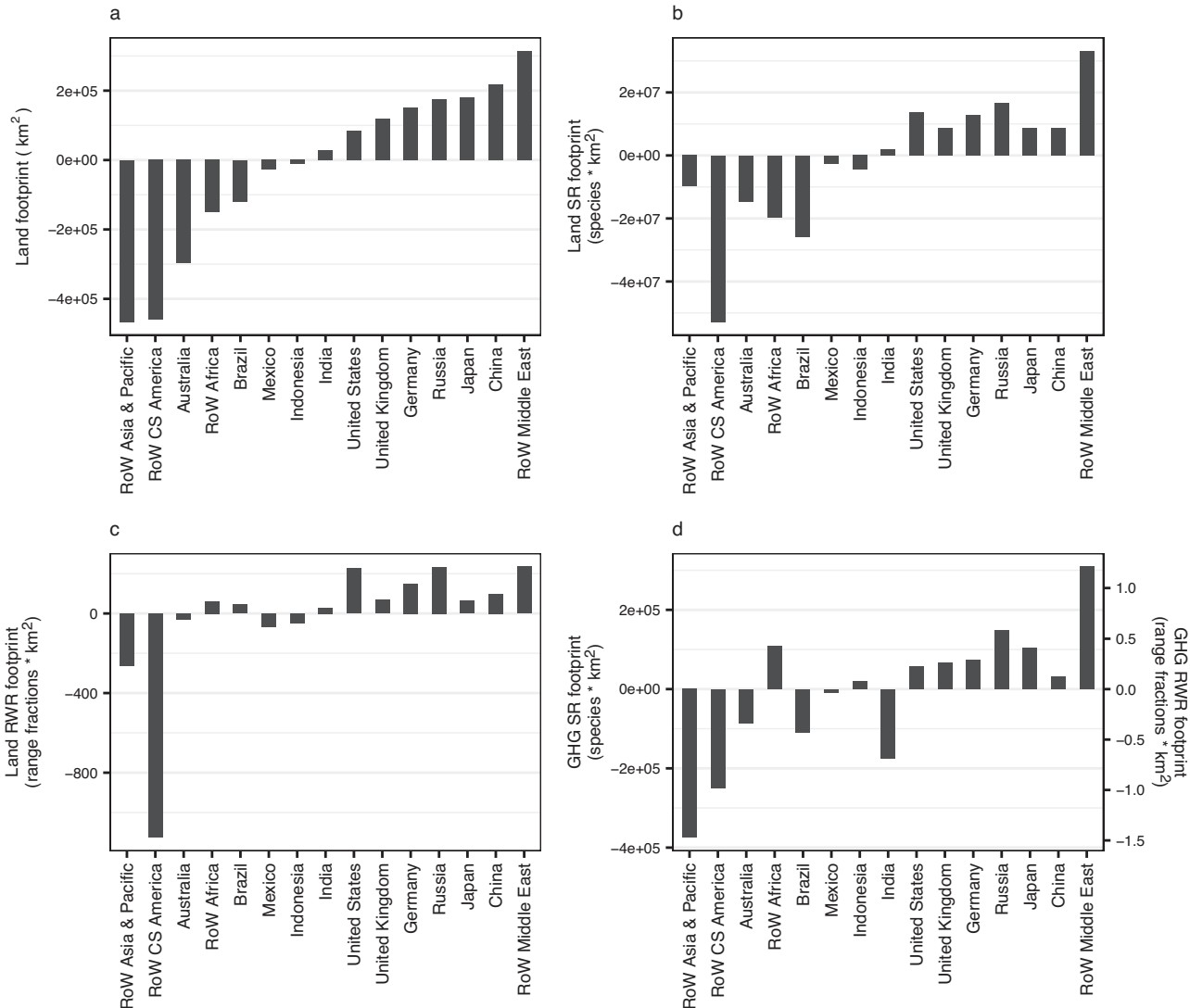

**Fig. 5 | Regions' net import footprints for food-related products for the year 2011.** Footprints are in terms of (**a**) land use, (**b**) land-driven species richness (SR) loss, (**c**) land-driven rarity-weighted richness (RWR) loss and (**d**) GHG-driven richness loss. Regions which are in the highest five (net importers) or lowest five (net exporters) for the net import footprints are shown. RoW Rest of World. Source data are provided as a source data file.

## Contribution of the different GHGs to biodiversity footprints

The total GHG-driven footprint of all food-related products is -1% of the total land-driven richness footprints. Food-related products accounted for nearly a quarter (23%) of the total emissions across all sectors of activity (including industry) in EXIOBASE in 2011. Methane emissions account for 70% of the total GHG-driven biodiversity footprint from food-related products, compared to 42% of the GHG-driven footprint of all EXIOBASE's products. Carbon dioxide contributes to 18% of food's total footprint versus 54% of the footprints of all products and nitrous oxide contributes to 12% of food's footprint and 4% of the footprints of all products.

Methane is the primary contributor to total food-related GHG-driven biodiversity footprint in all regions (Fig. 1d), with emissions from animal-based products, paddy rice and food waste being primarily from methane (Supplementary Fig. 5). In contrast nitrous oxide drives the biodiversity footprint for fertiliser products and 'fish and other fishing products' (Fig. 2d and Supplementary Fig. 5). The relative contributions of the different GHGs differs between regions, for example nitrous oxide contributing a larger proportion of the Netherlands' and Belgium's GHG-driven production-based footprint than Malta's (Fig. 3d) and carbon dioxide contributing relatively highly to

South Africa's consumption-based footprint but relatively little to Brazil's (Fig. 4d). At the product level, for example, wheat's GHG-driven footprint is primarily due to nitrous oxide in India but methane in China; nitrous oxide is the greatest contributor to China's fruit and vegetables' footprint but carbon dioxide to India's (Supplementary Fig. 5).

## Biodiversity loss embedded in export/import trade

The United States, the United Kingdom, Germany, Russia, Japan, China and RoW Middle East – regions that tend to be relatively biodiversity-poor and highly industrialised in their land use systems – are all net importers of land-driven biodiversity loss and also of GHG-driven biodiversity loss (Fig. 5, Supplementary Fig. 9). Once again, the different land-driven footprints also show different messages. For example, RoW CS America exports a similar land-area footprint to that exported by RoW Asia & Pacific but approximately four times RoW Asia & Pacific's exported rarity-weighted richness footprint. RoW Africa is a net exporter of land-driven species richness impacts but a net importer of land-driven rarity-weighted richness impacts. The three greatest net importers of both land-driven biodiversity footprints are RoW Middle East, Russia and the United States. Regions which are net importers of

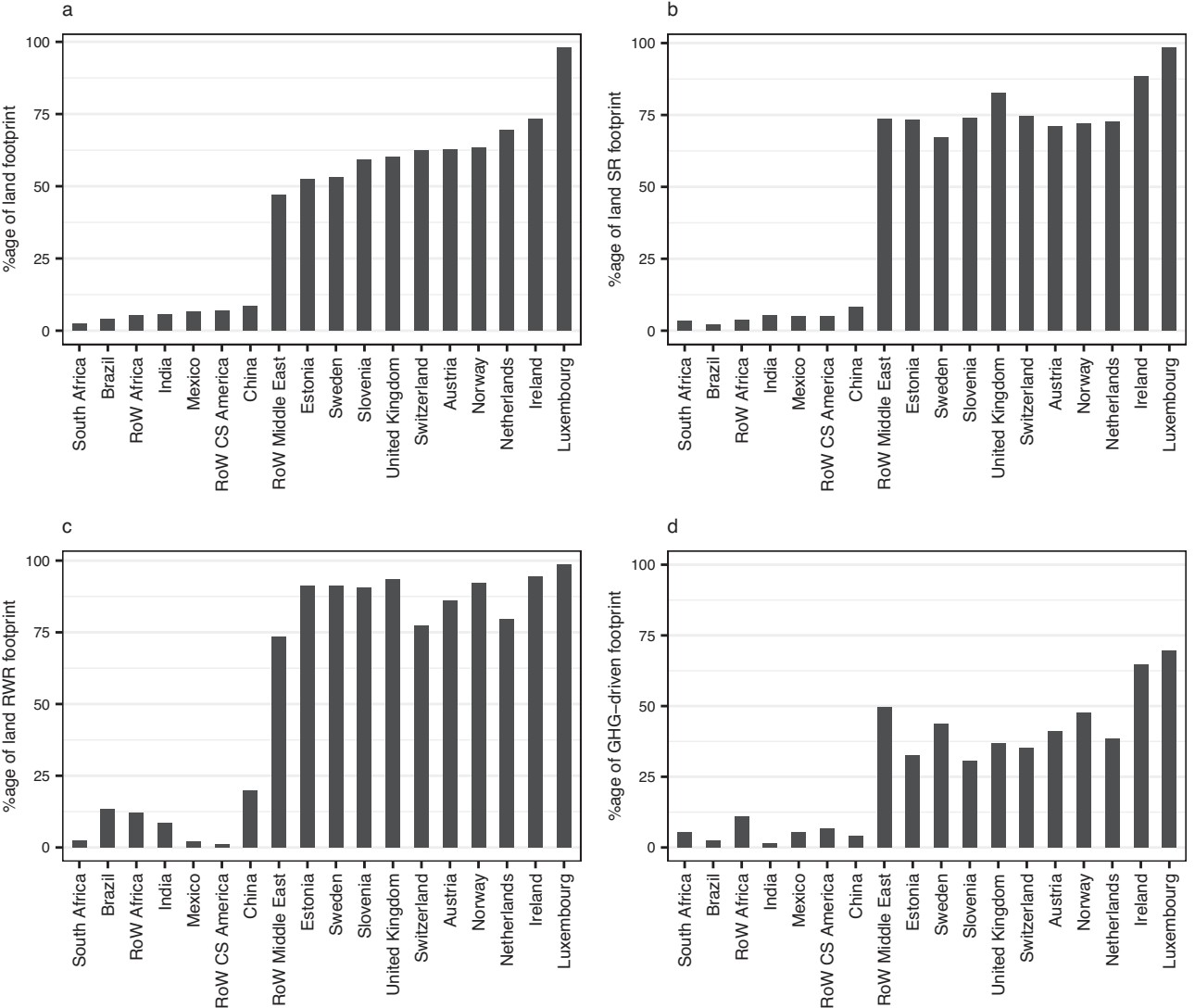

**Fig. 6 | The percentage of a region's food-related footprint that was imported in the year 2011.** Footprints are in terms of (**a**) land use, (**b**) land-driven species richness (SR) loss, (**c**) land-driven rarity-weighted richness (RWR) loss and (**d**) GHG-driven richness loss. Regions which are in the 5 lowest or highest percentages for one or more footprints are shown. RoW Rest of World. Source data are provided as a source data file.

land-driven biodiversity footprints also tend to be net importers of GHG-driven footprints with the exceptions of RoW Africa, Indonesia (net exporters of land-driven but importers of GHG-driven footprints) and India (net importer of land-driven but exporter of GHG-driven footprint).

Understanding the percentage of a region's footprint that is imported is important in devising policies to reduce footprints. Imports make up 5% or less of land-driven species richness footprints for highly biodiverse regions such as Brazil and Mexico (Fig. 6b, Supplementary Fig. 10); reducing domestic consumption in these regions might actually lower their consumption-based footprints. In contrast, the regions with the highest percentages of imported footprints are European. Luxembourg, the Republic of Ireland and the United Kingdom import 98%, 88% and 83% of their land-driven species richness footprints respectively. These high percentages will in part be due to the regions' relatively low domestic food production footprints but may also reflect imports grown in highly biodiverse regions, which could potentially be sourced more sustainably. Luxembourg, Ireland and Norway import high percentages of their GHG-driven biodiversity footprints (69%, 65% and 48% respectively) (Fig. 6d) and also have particularly high per capita consumption-based GHG-driven footprints (Fig. 4d).

We find that 10%, 15% and 8% respectively of land-driven species richness, land-driven rarity-weighted richness and GHG-driven richness footprints are embedded within trade between aggregated world regions (Supplementary Fig. 11), with patterns differing between the footprint types, and between animal and plant-derived foods (Supplementary Figs. 11-13). Although the total footprints of animal-derived products are higher than plant-derived, a greater percentage of the plant-derived footprint is traded between world regions: ~8%, 15%, 7% (animal-derived) versus 17%, 21%, 11% (plant-derived) of the total footprint for land-driven species richness, land-driven rarity-weighted richness and GHG-driven species richness, respectively. These results are discussed in more detail in the Supplementary Information (Supplementary Fig. 13).

## Discussion
Our metric of land-driven species richness change captures the higher biodiversity footprint of products from more species-rich regions and from biomes in which biodiversity is more sensitive to land-use change. The metric thus provides a more accurate footprint of the biodiversity impact embedded within food than would be obtained using land area alone. Weighting richness by species' range area further changes the assessment of embedded biodiversity footprint.

Rarity-weighted richness places a greater emphasis on the biodiversity costs in areas containing a higher proportion of narrow-ranged species, which are of greater conservation concern than wider-ranging species, notably parts of CS America. Our analysis also shows the need to consider GHG-driven biodiversity loss, particularly in assessing longer term future impacts of food production. We find that the climate contributions of food production are accumulating rapidly over time – in several regions, annual GHG-driven production footprints are as high as 2% of the total historic land-use footprint, meaning that just a decade's worth of emissions (not including land conversion emissions) will add an additional 20% to the biodiversity loss that has already occurred due to wholesale conversion of land to agriculture. 70% of the total GHG-driven biodiversity footprint of food-related products stems from methane.

Our study incorporated the combination of: (i) allowing sensitivity to land use to vary by land-use type and biome; (ii) allowing for the natural variation of species richness across political regions; and (iii) allowing for spatial variation in future temperature change and in species' responses to that temperature change. The models we used to capture land-use and climate impacts output the same biodiversity metrics, allowing us to compare these two major drivers of biodiversity change. Comparing the results of our study with those of previous analyses of the embedded biodiversity footprints of food is not straightforward since studies generally differ in resolution with respect to regions and/or products (and may use data from different years). Where comparisons can be made, we find both common ground and divergence. In line with Marques et al.[27], we show that production in Africa, CS America and Asia & Pacific regions has the highest overall biodiversity footprints and that cattle products have a particularly high impact on biodiversity globally. Marques et al.[27] use a biodiversity metric of number of impending extinctions of bird species and, in common with our land-driven species richness metric, identify almost equally high impacts of cattle in Africa as in CS America. However, it is only by using our rarity-weighted richness measure that the particularly high cost to narrow-ranged species from cattle products in CS America is revealed. Chaudhary and Kastner[16] use bilateral trade data in combination with a metric of the number of species committed to extinction. Their study had greater spatial disaggregation than ours, meaning we cannot compare regions included within our RoW regions. An extension of EXIOBASE disaggregates trade regions into 214 countries, but its environmental extension has fewer land use categories and does not cover GHG emissions[38]. Nevertheless, our results broadly support each other with respect to highlighting particularly high consumption footprints in India, China, Brazil and the US. However, Chaudhary and Kastner found Indonesia to have the second highest consumption footprint, whereas it only appears in our top ten consumption footprints for rarity-weighted richness. The relative magnitude of the regions with the greatest consumption footprints in Chaudhary et al.[39] study differs from our results. Sun et al.[40] also use a metric of the number of species committed to extinction at a greater spatial disaggregation than our study. Their results show strong agreement with ours with respect to consumption per capita, both studies finding high footprints in Central America, Brazil, South Africa and Australia. Kitzes et al.[17], using metrics based on birds and a greater spatial disaggregation than our study, also find particularly high impacts of bovine products and, in contrast to our results, of processed rice. Lenzen et al.[24] assess the biodiversity footprint, as measured by number of threats to species, of all commodities, not just those associated with food. Both Lenzen et al.[24] and our study find the US, Germany, the UK and Japan to be among the greatest importers of embedded biodiversity, but our study has Russia as a major importer of embedded biodiversity whereas Lenzen et al.[24] find it to be one of the greatest net exporters. The discrepancies between studies will in part result from differences in trade models[41,42] but are also likely to result from differences in the biodiversity metrics used[28], adding support for ours and Marquardt et al.'s.[28] findings that different biodiversity metrics lead to different conclusions.

Our estimates of the impacts of food-related GHGs on biodiversity are lower than those of Wilting et al.[29] which is at least in part due to differences in methodology. We predicted the temperature rise in 2031 due to an emissions pulse in 2011 using the global temperature potential (GTP). Wilting et al.[29] used the integrated GTP over a 100 year time horizon, thus summing the warming that occurred in every year following the emission pulse, as opposed to calculating the temperature of the hundredth year only. Wilting et al.[29] analyzed the land-driven and GHG-driven biodiversity loss from all economic sectors and estimated that GHGs contributed to an average of 18% of the biodiversity loss associated with food although contributions varied by region from 7% (Africa) to 45% (rest of Oceania). Given the different metric and time horizon of global warming used, we would expect our results to differ from Wilting et al.[29], with our GHG-driven footprints estimated to contribute from the order of 0.1% – 6% of biodiversity loss for food products and up to 16% for fertilizer. Wilting et al.'s[29] study further differed from ours since it used a different biodiversity metric and MRIO, and assumed agricultural land was evenly distributed across regions. Our studies support each other in suggesting current GHG-driven emissions from the food-related products will play a very significant role in future biodiversity loss. Reducing emissions associated with food is therefore a high priority, particularly in Europe, North America, Asia & Pacific.

Our study shows that measurements of trade-related impacts on biodiversity differ considerably depending on the biodiversity metric used. Whilst detailed discussion of the merits and relevance of different biodiversity metrics may be appropriate in the scientific literature, it is unlikely to be helpful to businesses and policymakers who require succinct and readily interpretable biodiversity footprints. This leads to the question of which measure(s) should be recommended to the public and private sectors to aid them in reducing their environmental footprints. What might the consequences be of one metric being chosen over others? Our study suggests that a metric of land area, whilst easy to measure and interpret, will over-inflate the cost of biodiversity in biodiversity-poor areas and under-estimate it in biodiversity-rich areas, and thus is not recommended when the focus is biodiversity conservation. Species richness is preferable in this case, and is still relatively easy to interpret, particularly if used in relative rather than absolute terms. However, species richness has been criticised as a biodiversity metric, since it does not decline if rare species are replaced with common species (see Hillebrand et al.[33]). Weighting for species rarity overcomes this, and highlights the higher losses of smaller-ranged species from CS America. Using rarity-weighted richness over a simple species richness measure would likely lead to different decisions with respect to the sustainable trade of food products.

We must also consider whether GHG-driven biodiversity loss adds value or unnecessary complexity to footprint estimates. Our study has shown that it is fairly simple to calculate GHG-driven biodiversity loss and to aggregate it with the land-driven impact. The annual GHG-driven biodiversity impact of livestock sectors is equivalent to ~5% of land-driven biodiversity loss in Asia, the US and Western Europe. Importantly, this equates to 50% of the biodiversity impact of complete land conversion within the timespan of just one decade. For some individual products the ratio of GHG to land-driven impacts is even higher. Including the GHG-driven losses in estimates of the food industry's impact on biodiversity would therefore seem worthwhile, particularly since the bulk of emissions are from methane. Methane's short lifetime means that (assuming methane sinks are constant), reducing methane emissions actually leads to global cooling but on the flipside, increased emissions lead to substantial rapid warming[43]. Reducing emissions from food in the immediate-term would therefore be an extremely effective route to reducing near-term warming. Our results add to the already weighty evidence, e.g., [11,44,45], in support of

policies that assist farmers to transition away from livestock and nudge consumers toward a more plant-based diet.

Scientists are increasingly called on to provide support to assess and reduce biodiversity impacts in both the public and private sectors[46,47]. The biodiversity footprint methodologies that we present could be used by multiple stakeholders in the global food system, e.g., informing supply chain management, decisions on dietary shifts and policies on sustainable trade, in particular, free trade agreement (FTA) chapters targeted at reducing biodiversity loss. The European 'new green deal' aims to improve the state of the environment in Europe but this will involve offshoring part of Europe's biodiversity footprint to the tropics[48]. Tropical farming systems tend to have lower yields[49], lower environmental regulations[48] and higher baseline biodiversity[50] than European systems, meaning that not only is more land required to produce the same amount of food but, as we show, the cost to biodiversity per unit area is also relatively higher. There are deep concerns that recent agreements such as the 2021 UK-Australia FTA and the 2019 EU-Mercosur FTA will increase tropical deforestation and biodiversity loss[51] via insufficient sustainability regulations. Our study shows that Australia and CS America already dominate the exported biodiversity footprint embedded in cattle products and that if this footprint is to be decreased, policymakers should be looking at reducing, not increasing, the export of such products. The metrics we have developed could easily be incorporated into online tools such as https://commodityfootprints.earth/ which provide highly accessible ways for stakeholders to understand the environmental impacts of current consumption patterns. It is important that such tools use metrics that capture different aspects of biodiversity since, if based on land use alone, impacts on small-ranged species in the tropics will be substantially underestimated, particularly in CS America.

The examination of production-based footprints per-area (which facilitate comparisons of production between different-sized regions) allows for an increased understanding of the implications of regional/national agricultural policies. Regions with high per-area production-based footprints are targets for policies that reduce the environmental impact of farming. There is considerable overlap of high per-area land-driven and GHG-driven production impacts and it is interesting that the Netherlands and Belgium, both less species-rich temperate countries, have some of the highest per-area production-based footprints, even for rarity-weighted richness (Fig. 3). Taiwan is notable for all three of its per-area land-driven production-based footprints being extremely high (Fig. 3), with impacts being driven by a variety of products including fruit/vegetables/nuts, other animal products (e.g., eggs, honey), fish, pork products, sugar, and other food products (e.g., pizza, soups, sauces) (Supplementary Fig. 7) (although see ref. 36 regarding possible over-estimation of Taiwan's material footprint). India stands out as having very high paddy rice production per area for all footprints (Supplementary Fig. 7).

Per-capita consumption-based footprints (which facilitate comparisons of consumption between regions with different population sizes) also allow for targeting of agricultural and trade policies. We found China, India and RoW Africa had the highest overall GHG-driven biodiversity footprints, but once population size was considered, only RoW Africa remained in the top ten per-capita footprints. Identifying the products that lead to regions' high per-capita footprints is the first step towards reducing those footprints. For example, extremely high per-capita consumption-based footprints from cattle, raw milk and dairy products contribute to Estonia's place in the top ten for GHG-driven biodiversity loss (see Fig. 4 and Supplementary Fig. 8). Whilst we recognise that changing cultural norms can be very difficult, reducing Estonia's per-capita footprint would be relatively straight forward from a policy perspective, since it involves only one food group and could perhaps be tackled via a combination of dietary shifts, source-shifting and change in dairy-farming practices. Australia, in contrast, has particularly high per-capita land-driven consumption-

based footprints for a wide variety of products spanning fruit/vegetables/nuts, all products associated with cattle and other meat animals (e.g., sheep), rice, other food products and beverages. Reducing these footprints will likely require a more complex suite of strategies.

We found that some regions import over 90% of their embedded biodiversity footprint, further indicating the need for sustainable trade as a measure to halt biodiversity loss. Regions with a high proportion of imported footprint coupled with high per-capita consumption-based footprints are targets for reducing their imports and/or the biodiversity footprint embedded within those imports[17]; Estonia, Luxembourg, the Netherlands and Norway all fall into this category (Figs. 4 and 7). Conversely, regions with a low percentage of imported footprint but high per-capita consumption such as Brazil, Mexico and South Africa should be lowering their domestic footprint in order to reduce their biodiversity impact.

The main net importers of land-driven biodiversity footprint tend to be net exporters of GHG-driven biodiversity footprint, and vice versa. The emissions associated with crop production largely arise from fertilizer use (although also fuel combustion, industry and waste)[52]. Low fertilizer application will be associated with low emissions but lead to a smaller yield and therefore a larger land-use footprint. On the other hand, heavy fertilizer use will result in more emissions but smaller land use. This negative correlation is exacerbated since areas that use more fertilizer tend to be areas of lower species richness such as N America and W Europe[53]. Future studies could use our comparable land-driven and GHG-driven biodiversity metrics to investigate the reduction of biodiversity footprints via source shifting and the spatial optimization of cropland.

There are several limitations of MRIOs which must be considered in interpreting our results: sectoral and regional aggregation[54,55], an assumption of price homogeneity[56,57], limitations in data reporting[58], and the age of input data[59]. Sectoral and regional aggregation are particularly pertinent given that some of the highest footprints we find relate to the highly aggregated RoW regions and 'Other food' sector. Regional aggregation underestimates the volume of trade (and thus embodied impacts) and has been shown to under/over estimate the land use impacts of agricultural commodities by up to 20% and 10% respectively, compared to a disaggregated version of EXIOBASE with full country resolution[38]. The original EXIOBASE 3 data end in 2011 and although more recent trade data are now available, land use is still limited to 2011 (https://zenodo.org/record/5589597#.YnkHvOjMK3A). Our results therefore reflect the biodiversity impact of food production over a decade ago, and will likely underestimate the current impact due to the expansion of agricultural land[60] in the intervening period.

Agricultural intensity is not yet captured within MRIO models of trade flows. Consequently, in common with other footprinting studies, e.g., [26,27], our analysis only measures biodiversity impacts caused by direct land-use change, and does not explicitly consider the impacts of agricultural intensification. Neither does our study explicitly account for indirect biodiversity loss that might occur through loss of ecosystem function, e.g., soil impoverishment. Nevertheless, the PREDICTS database does sample gradients of agricultural intensity and ecosystem degradation, and so these factors should be captured implicitly[31]. New approaches, such as the Human Appropriation of Net Primary Production (HANPP), e.g., [21], show promise for capturing effects of agricultural intensity in the future, although linking such measures to biodiversity loss remains a challenge. Our models of land-use impacts represent terrestrial vertebrates, invertebrates, plants and fungi but, due to data limitations, our biodiversity metrics used to estimate climate impacts refer to terrestrial vertebrates only.

While our study highlights that the consumption of cattle products contributes considerably to land-driven biodiversity loss, it is important to be aware that there is high variability in pasture maps stemming from the different definitions, methods and underlying

datasets used in their generation[61]. However, the Ramankutty et al.[62] dataset that we used is one of the best available, aligning well with other datasets[61].

We have presented methods that capture regional variation in the sensitivity of biodiversity both to land use and climate change to estimate the land-driven and GHG-driven biodiversity impacts embedded within the production, consumption and trade of food-related commodities. We find that land-driven footprints differ depending on the metric used and that GHG-driven biodiversity impacts are driven largely by methane emissions and contribute a higher proportion of the total footprint in regions of higher income and lower species richness. The measures we present are simple to calculate and could be incorporated into decision-making and environmental impact assessments by governments and businesses. Our consistent metric for biodiversity impacts allows us to present multiple aspects of land-driven and GHG-driven biodiversity footprints, enabling insight into priorities for reducing biodiversity costs via both global and regional production and consumption.

## Methods
### EEMRIO analysis
Input-output analysis is a top-down approach that uses sectoral transaction data (either in financial terms or units of product) to account for the complex interdependencies of industries. An input-output table can be environmentally extended by adding information on exchanges with the environment, e.g., GHG emissions or land use, by each industry sector. EEMRIO analysis traces the production and supply chain of traded goods and services and their associated materials back to the source of primary extraction, thus capturing the direct and indirect environmental pressures associated with a country's final consumption (from households, non-profit organizations, governments, capital formation, changes in inventories and valuables, and exports)[20]. Full details of the calculations underlying EEMRIO analysis can be found in Kitzes[20] and Miller and Blair[19].

We used the standard environmentally-extended Leontief model to calculate the effects of consumption on biodiversity loss

$$\mathbf{E} = \mathbf{f} \cdot (\mathbf{I} - \mathbf{A})^{-1} \cdot \mathbf{Y} \tag{1}$$

Where, for $i$ regions and $m$ economic sectors:

$\mathbf{E}$ is a $(1 \times i)$ matrix, representing the environmental impacts associated with the final demand of each region ($CO_2$, $CH_4$ and $N_2O$ emissions, in kg, and agricultural area, in $km^2$).

$\mathbf{f}$ is the $(1 \times im)$ direct intensity vector, representing the environmental pressures (e.g., area of land, mass of $CO_2$ emissions) associated with one unit (€1 M) of production for each product sector in each region.

$\mathbf{I}$ is the $(im \times im)$ identity matrix.

$\mathbf{A}$ is the $(im \times im)$ technical coefficient matrix which gives the amount of input (€1 M) that every sector must receive from every other sector in order to produce one unit (€1 M) of output.

$\mathbf{Y}$ is the $(im \times i)$ matrix of final demand (associated with households, non-profit organisations, governments, capital formation, changes in inventories and valuables and exports) given in monetary terms (€1 M).

The direct intensity vector $\mathbf{f}$ allows the MRIO to be extended to include environmental costs of production and consumption. Sections 2.2 and 2.3 describe our environmental 'characterization factors' based on species richness, rarity-weighted richness, land use, and GHG emissions. The vector relating to land use is a sparse vector populated in the entries for production activities that directly involve use of cropland or pasture, e.g., populated for paddy rice but not for processed rice.

We use the EEMRIO database EXIOBASE3.8.1 (product by product version)[52], because of its superior balance of sectoral and regional disaggregation relative to other MRIOs. EXIOBASE3 provides a harmonised time-series of MRIO tables and environmental extensions over the period 1995-2022, covering 200 products, 44 countries (EU countries and other major economies), plus 5 'Rest of World' (RoW) regions. We chose 2011 as our year for analysis since land use data are available up to 2011 only (https://zenodo.org/record/5589597#.YnkHvOjMK3A) and spatial data for the majority of crops[49,63] does not extend beyond this date (see below).

Our analysis focuses on the production and consumption of the 33 product sectors associated with food (Supplementary Table 1) in the year 2011, including fertilizer production and food waste processing. 'Hotel and restaurant services' were excluded since they encompass energy and materials, for example, as well as food. Full details of EXIOBASE's land-use and emissions accounts are given in Stadler et al.[52]. Our land-driven biodiversity footprints are based on agricultural land, and exclude impacts of forestry, infrastructure or other uses. We account for land use intensity in the sense that the amount of land required to produce one unit of food differs between the EXIOBASE regions but, in common with previous biodiversity footprint studies, we do not model the effects of different agricultural practices (e.g., fertilizer and pesticide use) or different yields per unit area on biodiversity. Direct GHG emissions from crops derive from nitrogen fertilisers, while direct GHG emissions from livestock derive from the animals (enteric fermentation for ruminants), the manure excreted and the cultivation of feed crops. Impacts of fertilizer refer to the production of the fertilizer only, and impacts of food waste refer to the waste treatment and decomposition processes and do not include the production impacts of the wasted food.

### Characterisation factors for land-driven biodiversity impacts
Maps for a) the 49 EXIOBASE trade regions, b) the six biome groupings used in Newbold et al.[31] (but here also including boreal forest within the temperate-forest grouping), and c) the 15 EXIOBASE agricultural land-use categories were identified, and map masks for each unique 3-way combination were generated in R using packages raster[64] and rgdal[65]. Shapefiles mapping the borders of individual countries[66] were aggregated and rasterized to match EXIOBASE's 49 regions. Biomes were obtained from The Nature Conservancy[67] and were aggregated into tropical forest, temperate/boreal forest, tropical grassland, temperate/montane grassland, Mediterranean, and drylands (tundra, mangroves, flooded grasslands, inland water and rock & ice excluded; Supplementary Table 2). Cropland data were obtained from the Spatial Production Allocation Model (SPAM)[49] and from EarthStat[63] and pasture data from EarthStat[62]. All spatial data were reprojected to an equal-area Behrmann grid. A summary schematic is given in Fig. 7a and further details of the preparation of datasets are given in the Supplementary Information.

Mapped spatial estimates of species richness and rarity-weighted richness for terrestrial vertebrates were obtained by stacking species geographical distributions at a 10-km equal-area resolution. Extent-of-occurrence distribution maps were obtained from the IUCN[68], BirdLife International[69], Meiri et al.[70], and Roll et al.[71]. We selected only areas where species are extant or probably extant, and resident during the breeding season. As in Etard et al.[72], we further excluded areas outside the known elevational limits of species[68]. Species richness was calculated as the number of species occurring within a grid cell. Rarity-weighted species richness weights were calculated as the inverse of a species' estimated range size, and rarity-weighted richness as the sum of these weights across species occurring within a grid cell. Mean species richness and rarity-weighted richness were calculated for each land-use type within each biome within each trade region.

We estimated the sensitivity of biodiversity to land use using models of the PREDICTS database[73,74], following the methods given in Newbold et al.[31] (for full details see the Supplementary Information).

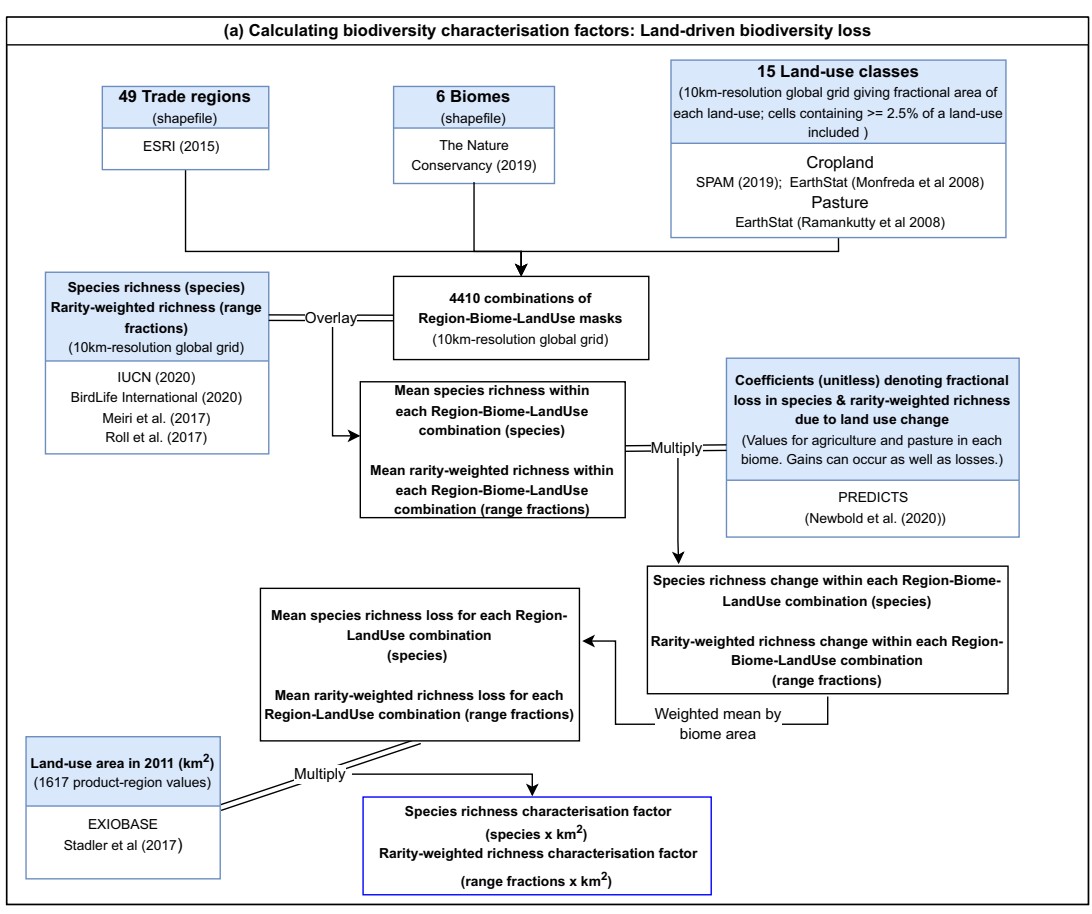

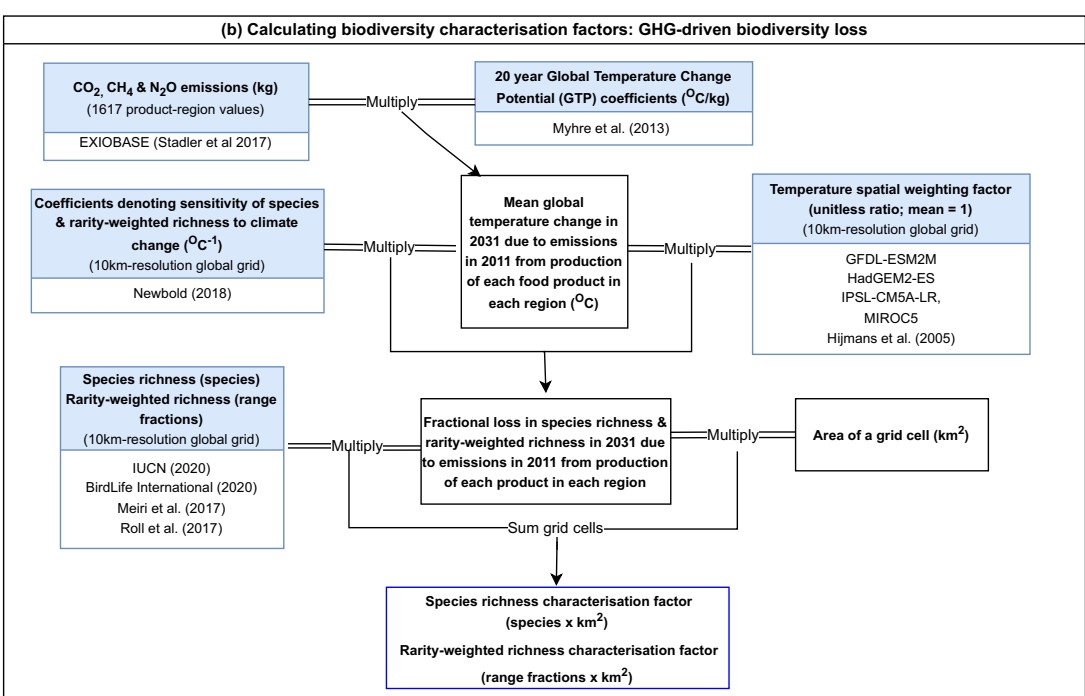

**Fig. 7 | Schematics summarising the characterisation factor calculations.** Characterization factors for (**a**) land-driven biodiversity loss and (**b**) GHG-driven biodiversity loss. The methods allow for biodiversity gain which would be represented as a negative loss.

The PREDICTS database contains 3.25 million samples of biodiversity in six different land uses from 26,114 locations and 47,044 different species. For the purposes of this study, we modeled biodiversity differences among four land-use classes: primary vegetation, secondary vegetation, agriculture (including both plantations and herbaceous cropland) and pasture, allowing the response of biodiversity (i.e., species richness based on 21,986 locations from 637 studies and rarity-weighted richness based on 15,198 locations from 403 studies) to these land-use categories to vary among the six broad biome groupings listed above.

Characterization factors for species richness were calculated as follows (see also Fig. 1a). We used the species richness map to estimate the mean species richness $S_{i,j,k}$ that would occur in undisturbed land across the area covered by each EXIOBASE land-use type $i$, within each biome $j$ and each EXIOBASE trade region $k$. The change in species richness $\Delta S_{i,j,k}$ due to land use was calculated as:

$$\Delta S_{i,j,k} = S_{i,j,k} \times P_{j,l} \qquad (2)$$

where $P_{j,l}$ is the proportional change in species richness in PREDICTS land-use type $l$ and biome $j$, compared to primary vegetation in that biome (see Supplementary Table 3 for the correspondence between EXIOBASE and PREDICTS land-use types, and Supplementary Tables 4 and 5 for values of $P_{j,l}$). The change in species richness $\Delta S_{i,k}$ within each EXIOBASE land-use type $i$ and each region $k$ was calculated as:

$$\Delta S_{i,k} = \sum_{j=1}^{J_{i,k}} (\Delta S_{i,j,k} \times B_{i,j,k}) \qquad (3)$$

where $J_{l,k}$ is the number of biomes covered by the EXIOBASE land use type $i$ within the region $k$ and $B_{i,j,k}$ is the proportion of the EXIOBASE land-use type $i$ within the region $k$ that is covered by the biome $j$. The characterisation factor $CF_{i,k}$ was then calculated as:

$$CF_{i,k} = \Delta S_{i,k} \times A_{i,k} \qquad (4)$$

where $A_{i,k}$ is the area of agricultural land used to produce €1 M of product in land-use type $i$ in region $k$, as given by EXIOBASE). Characterization factors for rarity-weighted richness were calculated using the same method, but using the map of rarity-weighted species richness, and corresponding modelled estimates of land-use sensitivity (for characterization factor values, see Supplementary Data 1 and 2).

The characterization factors for species richness have units of number of species × km$^2$ and can be thought of as the count of the species lost, with this loss extending over the area of land required to produce €1 M of product, as given in EXIOBASE. Rarity-weighted richness characterization factors are less intuitive but can be thought of as having a unit of species range fractions × km$^2$.

**Characterisation factors for GHG-driven biodiversity change**

We calculated GHG-driven biodiversity change in a three-step process. First, we calculated the GHG emissions (split into carbon dioxide ($CO_2$), methane ($CH_4$) and nitrous oxide ($N_2O$)) associated with each region's production of the 33 different food-related products in 2011 using EXIOBASE. Second, we calculated the warming that would result from these emissions. Third, we calculated the biodiversity change that would result from this warming, allowing for non-uniformity of warming across the globe. A summary schematic is given in Fig. 7b.

EXIOBASE gives emissions for six GHGs: carbon dioxide, methane, nitrous oxide, and three fluorinated gases. We calculated footprints using the first three gases only since their contribution to warming via food-related emissions is much greater than that of the fluorinated gases[75]. Emissions were summed over the four categories given by EXIOBASE ('Combustion – air', 'non-combustion', 'agriculture' and 'waste'). Footprints were calculated using the environmentally-extended Leontief model described above.

GHG-induced warming is often described in terms of the Global Warming Potential (GWP), a metric that compares the radiative forcing integrated over a time period caused by the emission of 1 kg of an agent relative to the integrated forcing caused by the emissions of 1 kg of $CO_2$[76]. However, GWP has been criticised as it does not translate into a climate response that is intuitively understood[77]. We, therefore, calculated the increase in global surface temperature due to GHG emissions using the Global Temperature Change Potential (GTP), a metric designed to be an intuitive measure of climate response[77] and one that has been used in previous biodiversity footprint studies[29]. The GTP is defined as the ratio between the global mean surface temperature change at a given future time horizon following an emission (pulse or sustained) of a compound relative to a reference gas[77]. Studies often use a time horizon of 100 years following the Kyoto Protocol, but this choice was originally made on an arbitrary basis and is not the most appropriate for shorter term continental climate responses as it masks methane's potency[11,78]. Instead we chose a 20-year time horizon, to capture warming due to the relatively short-lived methane emissions, and to represent a time that is tangible to today's policy and decision-makers. GTP values in units of degrees of warming (°C) per kilogram of emissions were taken from the AR5 IPCC report[79], Table 8.A.1 ($CO_2$ $6.84 \times 10^{-16}$ °C /kg, $CH_4$ $4.62 \times 10^{-14}$ °C /kg, $N_2O$ $1.89 \times 10^{-13}$ °C /kg). The warming in °C that would result by 2031 $\Delta T_{p,k}$ due to the emissions $E_{g,p,k}$ associated with each gas $g$ due to the production of €1 M of each product $p$ in each region $k$ was calculated as:

$$\Delta T_{p,k} = \sum_{g} (E_{g,p,k} \times C_g) \qquad (5)$$

where $C_g$ is the GTP value for each gas $g$.

We estimated the sensitivity of biodiversity to climate change based on future projections of changes in the distributions of terrestrial vertebrates under the Representative Concentration Pathways (RCP) climate-change scenarios[5], thus putting our projected temperature change in context with wider climate variables. It is necessary to use future projections rather than observed responses of species to climate change because we do not yet have enough data for a wide range of species from locations that have experienced extensive historical climate change[31]. Full details are given in the Supplementary Information.

To account for the non-uniformity of warming across the globe, we calculated a grid of spatial weighting factors to up or downweight temperature change depending on the difference between the projected local and mean global temperature change. The projected temperature change $\Delta T_x$ in each cell $x$ between 2011 and 2031 under RCP8.5 was calculated by averaging the projections of four different climate models (GFDL-ESM2M, HadGEM2-ES, IPSL-CM5A-LR, MIROC5[80]). The temperature weighting factor $F_x$ for each cell $x$ was calculated as

$$F_x = \Delta T_x / \Delta \bar{T}_x \qquad (6)$$

where $\Delta \bar{T}_x$ is the mean global temperature change. The fractional loss in species richness $\Delta S_{x,p,k}$ in 2031 due to $\Delta T_{p,k}$, the temperature increase from emissions from food production of each product $p$ in each region $k$, was calculated as:

$$\Delta S_{x,p,k} = F_x \times H_x \times \Delta T_{p,k} \qquad (7)$$

where $H_x$ is a grid estimating the sensitivity of species richness to climate change (for details see the Supplementary Information). This grid is based on projected changes in the distributions of species under climate change and describes the expected change in local species richness in any terrestrial location associated with a temperature increase of 1 °C.

This fractional loss in species richness $\Delta S_{x,p,k}$ was then multiplied by our species richness grid and by the area of each grid cell to give a characterization factor for GHG-driven biodiversity loss, $CF_{p,k}$, with a unit of species × km$^2$ that is comparable to our metric for land-driven biodiversity loss. This characterization factor was calculated as:

$$CF_{p,k} = \Delta S_{x,p,k} \times S_x \times A_x \qquad (8)$$

where $S_x$ is the species richness in undisturbed vegetation under a natural climate in grid cell $x$ and $A_x$ is the area of grid cell $x$. The same basic method was used to calculate the characterization factors for change in rarity-weighted richness (see Supplementary Information for details). The GHG-driven characterization factors for change in species and rarity-weighted richness are directly correlated (for characterization factor values, see Supplementary Data 3 and 4.)

Global maps illustrating the proportional richness changes that would occur due to land-driven and GHG-driven processes are given in Supplementary Fig. 4. Our metrics of land-driven and GHG-driven biodiversity loss are modeled and presented in the same units, and hence are comparable. However, there are differences in the way that the two drivers impact biodiversity, and in the methods we used to model these impacts. We calculate the biodiversity change associated with all land used in food production in 2011, regardless of the year of conversion. Our measure of GHG-driven biodiversity change is associated with emissions produced in 2011. We consider the impact of land use to be reversible and view land conversion as a 'one-off' cost, i.e., once the land is converted, biodiversity change is immediate and does not increase over time. In contrast, we view GHG emissions as irreversible, repeated annual costs that occur 20 years after emission. We would expect the global biodiversity loss caused by a single year of emissions to be much lower than that caused by the conversion of the total amount of agricultural land used in 2011. We calculate the ratio of land-driven to GHG-driven biodiversity loss for products and regions. We can crudely approximate a ratio of 100, for example, to mean that within a century, assuming the same emissions repeat annually, the global biodiversity loss caused by emissions from a region's food production will equal the biodiversity loss due to total land conversion in that region. In reality, this may be an underestimate since we do not consider the emissions released by land conversion.

### Per land area and per capita footprints

The regions within EXIOBASE have unequal land areas and human populations. To make a fairer comparison of footprints between regions, we converted production footprints to footprints per km² and consumption footprints to footprints per capita. Per-km² footprints were calculated using EXIOBASE trade regions' areas (Supplementary Data 5), which were calculated in R (using packages raster[64] and rgdal[65]) from ESRI country shapefiles[66]. Per-capita footprints were calculated using population data for the year 2011 for the EXIOBASE trade regions (Supplementary Data 6), which were obtained from CountryEconomy.com[81] (Taiwan) and the World Bank[37] (all other regions).

### Reporting summary

Further information on research design is available in the Nature Portfolio Reporting Summary linked to this article.

### Data availability

EXIOBASE version 3.8.1 can be downloaded from https://zenodo.org/records/4588235[82]. SPAM's Global Spatially-Disaggregated Crop Production Statistics Data for 2010 Version 2.0[49] can be downloaded from https://doi.org/10.7910/DVN/PRFF8V. EarthStat's Cropland[63] and Pasture Area[62] in 2000 can be downloaded from earthstat.org. Biome data from The Nature Conservancy's Terrestrial Ecoregions of the World 2009 data[67] can be downloaded from https://tnc.maps.arcgis.com/home/item.html?id=7b7fb9d945544d41b3e7a91494c42930. Country shape files[66] can be downloaded from https://hub.arcgis.com/datasets/a21fdb46d23e4ef896f31475217cbb08_1/data. GTP values can be found in the AR5 IPCC report[79], Table 8.A.1. Population data for the year 2011 can be downloaded from CountryEconomy.com[81] and the World Bank[37]. Species occurrence maps, the averaged temperature anomaly data, the cropland and pasture rasters and the species richness and rarity-weighted richness rasters used in the analysis are available on request from the authors. The compiled data underlying our study can be found in the Supplementary Data files 1–6. All other data supporting the findings of this study are provided in the Supplementary Information and Source Data files. Source data are provided in this paper.

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

## Acknowledgements

This work was supported by the UK Global Challenges Research Fund Trade, Development and Environment Hub (grant number ES/S008160/1) to TN; the UK Natural Environment Research Council (grant number NE/R010811/1) to TN and CD; by a Royal Society University Research Fellowship (grant number UF150526) to TN, by a UK Natural Environment Research Council Fellowship (grant number NE/N01524X/1) to CD and by a European Research Council Starting Grant (ERC, FLORA, 101039402, views and opinions expressed are however those of the authors only and do not necessarily reflect those of the European Union or the European Research Council Executive Agency) to CD. We are grateful to Simon Croft, Jonathan Green and Aafke Schipper for their advice regarding data analysis.

## Author contributions

E.B.: Conceptualisation, Methodology, Software, Formal analysis, Writing – original draft. C.D.: Writing – review and editing, Supervision. A.E.: Resources, Writing – review and editing. T.N.: Conceptualisation, Methodology, Software, Resources, Writing – review and editing, Supervision.

## Competing interests

The authors declare no competing interests.
