## [Peer Review File · Nature Communications]

Impacts of the global food system on terrestrial biodiversity from land use and climate changeREVIEWER COMMENTS

Reviewer #1 (Remarks to the Author):

This manuscript presents the analysis of food-related biodiversity footprints, it considers both land-driven footprints and GHG-driven footprints and two biodiversity metrics. The manuscript is very well-written, clearly structured and with high quality figures. However, I think it does not add sufficient novelty in its methods and findings. Therefore I cannot recommend this manuscript for publication in Nature Communications.

I hope the authors find my comments below useful.

General comments

My main issue with the manuscript in its current format is that it does not add sufficient novelty to the current state of the art. Marquardt et al. 2019 compared different biodiversity footprint metrics between them and also with land footprint. However the authors of this manuscript do not cite this work

(<https://www.sciencedirect.com/science/article/abs/pii/S1470160X19302687?via=ihub>).

Wilting et al. 2017 quantified land-driven and GHG-driven biodiversity footprints. However their study is more complete since they cover more sectors in their GHG-driven footprints (<https://pubs.acs.org/doi/10.1021/acs.est.6b05296>).

I have some questions regarding the methods used that need to be clarified (see Specific comments).

In the policy implications section it would be interesting to have a more targeted and in depth discussion on the application of the work to the current policy frameworks. This would add novelty to the work, a lot of developments in policy arena have happened since the publication of other biodiversity footprint related work.

Specific comments:

Line 145 – 147 – not clear which impacts refer to food waste. Can you please clarify?

Line 147 – 149 – you mention that “we regard land use as a one-off cost that results in a change in biodiversity and we assume that once land has been converted, it can be used repeatedly without the biodiversity cost increasing”. I don’t understand how can the impact be a one-off but then calculated for an yearly economic flow. Does this mean that if you

would calculate the impacts for 2012, there will be almost no impacts since they were attributed to 2011? I don't understand this sentence, could you please clarify?

Line 174 – nice figures, it would be more clear if you add the units in all squares so one can follow. It would make it easier to understand the final unit and how it is used.

Line 189 – 190 – Where all of these data points/studies used? I suppose you filtered the PREDICTS database to select only the metrics relevant to you, and a selection of studies with primary vegetation and another land use type. Please specify how many data points were used in the end. In the SI you should also identify this per model. It would be also nice to add a map with the geographical distribution of the studies used.

Line 197 – do you follow an approach presented in any other biodiversity footprint paper? If yes please refer to it in this section. If not, can you explain what are the novelties and differences in your approach?

Line 216 – not clear why and how you determine the amount of area needed to produce €1M of product.

Line 221 – I don't understand the units. Normally, the characterization factor represents the impact per unit of pressure (species lost/km² of land use). If the unit of the CF is species x km² when you multiply it by km² (from EXIOBASE) a different unit is achieved. This needs to be clarified.

Line 228 – Why final demand? Shouldn't you compute the emissions associated with production activities so that it is compatible with the environmental extension.

Line 238 – You choose to use Global Temperature Change Potential instead of the most commonly used approach using Global Warming Potential. Why is that the case? Please justify why you do not use GWP. I would also recommend calculating the impacts using GWP for two main reasons, first is that it is the most commonly used metric, which would increase the comparability of your results and second as a sensitivity analysis.

Line 241 – The choice of 20 years period should be better justified. If methane accounts for only a small percentage of the total gases I would argue that you should choose a different time horizon.

Line 251 – is the use of RCPs consistent with the use of GTP?

Line 255 – In the SI file you mention that "Species' projected responses to climate change were derived from¹⁰". What was the exact relationship? Can you add this information in SI?

Line 265 – 266 – you mentioned before emissions from final demand, but here you mention

emissions from food production of each product p in each region k . It is not clear what emission were used, please clarify.

Line 269 – please provide more information on the relationship used.

Line 272 – please clarify the unit (see comment referring to line 221)

Line 285 – I would say that considering land use as a one-off impact is not compatible with the analysis of the economic flows of one year.

Line 343 – Does this mean that the countries in the top 10 positions are the same for all the different metrics analyzed?

Line 370 – 372 – Why does a ratio of 100 indicates that reducing GHG biodiversity footprint is a priority?

Line 390 – I wonder if this is a real result of somewhat related with uncertainties of the MRIO table. Please check this papers to better understand if this is the case for this result:

<https://onlinelibrary.wiley.com/doi/10.1111/jiec.12833>

<https://journalofeconomicstructures.springeropen.com/articles/10.1186/s40008-020-0182-y#Sec20>

Line 418 -420 – I would expect Brazil to have a high per capita rarity-weighted footprint.

Why is that not the case? Can you please explain further?

Line 470 – 471 – If I am not mistaken in all figures for all the footprint the top 10 or 5 countries with highest footprints are the same, irrespective of the type of footprint. If this is true, doesn't it make it less important which footprint to use?

Line 497 – You don't mention the following studies:

<https://www.sciencedirect.com/science/article/abs/pii/S1470160X19302687>

<https://www.sciencedirect.com/science/article/abs/pii/S1470160X19302687>

<https://conbio.onlinelibrary.wiley.com/doi/full/10.1111/con4.12321>

Line 512 – 518 – These conclusions are misleading since Chaudhary and Kastner did not use an MRIO model which might have more influence on the results than the biodiversity metric used. Please revise this.

Line 519 - %21 – I don't think you can state this without a proper analysis. Also check the literature on the differences of the results obtained with physical and monetary approaches.

For example:

<https://www.sciencedirect.com/science/article/abs/pii/S0921800913003583>

<https://www.sciencedirect.com/science/article/abs/pii/S0921800915000932>

Line 530 – 536 – See my previous comments on the methods. I suggest you follow the same approach (at least as sensitivity) so that the comparison is possible.

Line 566 – I think you cannot really make this statement since you focus only on food products.

Reviewer #2 (Remarks to the Author):

I reviewed the analysis title „Impacts of global food supply on biodiversity via land use and climate change”.

The submitted manuscript investigates new methods by combining different datasets and using an EEMRIO to calculate land- and GHG-driven biodiversity footprints globally. Overall, the analysis is interesting, novel, and well-structured, however, there are a couple of issues and open questions that need to be addressed.

Line 46: But how climate change affects biodiversity globally is regionally still different, right?! In some regions with average colder temperatures, like Canada or Siberia, there might be more biodiversity evolving due to climate change?!

Line 49: Either here or later at line 89 you should differentiate and describe production vs consumption footprints.

Lines 86-88: For clarification, it would be helpful for the reader if you could state that your data are building on spatially explicit information but is aggregated and averaged to reflect heterogeneity but, in the end, do not represent spatially explicit information any more.

Lines 95-97: Do you refer to both production and consumption footprints here?

Line 117: When using Y from EXIBOASE the matrix is $7 \times i$, right? Because there are 7 different final demand categories. However, not sure if you need to mention that when you aggregate into one final demand category for each region. You could say “associated with the final demand of households, governments, capital formation, ... of each region.”

Line 130: “spare vector populated in the entries for ...” -- Sounds a bit odd. It shows which sectors/production activities require direct land-use. The same allocation principle applies to GHGs, right?

Lines 136-137: I understand, but I still have to flag this, it is quite old data and I’m unsure if 12-year-old data represent a proper fit to present data, also since you derive policy implications and not only methodological proof of concept. There are no land-use or spatial

crop data available after 2011?

Lines 147-149: How do you do that? A share of the upstream land-use flows based on spatial explicit information of land-use?

Figure 1a): So, for each biome you calculated a mean species richness/rarity richness and multiplied by the total land-use area reported in EXIOBASE?

Lines 177-178: You are only including animals, no plants. Would be good to mention somewhere.

Lines 227-229: How did you calculate “GHG emissions associated with each region’s 2011 final demand for the 33 different food-related products.”? Which method?

Lines 232-236: Again, what is the method for this – the formular? How did you calculate the GHG footprints for 33 different products? Output multiplier?

Lines 232-233: You don’t mention the expression of CH₄ and N₂O in CO₂ equivalents.

Lines 235-236: Also the N₂O from combustion and agriculture?

Lines 240: And at a given space/biome etc?

Lines 241: Given that you use data from the year 2011 (provided that they appear not too old for publication) you could check how the GTP model captured the actual changes in global temperature change; since 12 years of your 20-year time horizon have already passed.

Lines 285-286: Except for land use change that occurred in 2011 or the year before?

But how does this consideration affect your results and how you allocate emissions from land use to products?

Lines 306-307: You didn’t yet explain what the difference is and what they show (or I missed it).

Lines 321: Could you provide the total values for the GHG footprints of all food-related products, globally and/or nationally? This number could be compared to the numbers mentioned in the introduction in lines 31-32.

Lines 333-334: So, with their food system production they harm their domestic biodiversity as well as global biodiversity (GHG-driven). This could be mentioned to illustrate and differentiate.

Lines 342-343: I still don’t know the difference between production and consumption footprints, but they appear remarkably close to one another. That is not production- vs consumption-based approaches?!

Also, I realize these are total values, but I would recommend to at least mention why there are no European countries shown nor discussed.

Lines 359-361: I don't see this in e) or f). Isn't it Africa and C&S America that stand out? Or where can I find that information?

Lines 440-442: If correct, you could state here, again for illustration, that these countries are relatively biodiversity poor but highly industrialized in their land use system.

It would be useful to also mention how net-trade is calculated and what negative and positive values represent.

Lines 485-486: However, land-use intensity is still not reflected (probably early on in the manuscript). You should at least mention this and that other indicators, like HANPP (and embodied HANPP) can account for that.

Lines: 502-503: Actually, it would be interesting to also see the aggregated effects of both in a figure and to see it discussed in the text.

Line 514: You should not that there is an EXIOBASE version with higher national detail (n=214) but only captures land use as an environmental extension so far, not GHGs (see <https://zenodo.org/record/2654460>).

Lines 550-551: And method, or resolution respectively.

Lines 625-626: I'm not sure yet how you calculated the GHG footprints of food-related sectors/products. But since it's a supply-chain perspective, also machinery that is required by food producing sectors would fall into that, right?

Lines 626-628: It would be interesting to see (from a comparison) if it was useful from a biodiversity footprint perspective to onshore production to N America and W Europe from other regions to reduce land-driven biodiversity footprints and increase GHG-driven biodiv footprints but still decrease the total biodiv footprint by regionalizing production for consumption.

REVIEWER COMMENTS

We thank the reviewers very much for their comments and have addressed them all, as detailed below. We have paid particular attention to clarifying the methodology and we have also improved the policy implications section (see details below). Following our revisions, we believe the manuscript is much improved. In addressing the revision regarding the total carbon footprint we came across a bug in our code relating to the calculation of the GHG-driven footprint which had led us to underestimate this footprint. The main message of the results has not changed although the ratio of land-driven:GHG-driven footprints has decreased and we have updated the results, figures and discussion accordingly. In making the updates we realised that actually it would be interesting to split out the GHG-driven footprint into the separate contributions made by carbon dioxide, methane and nitrous oxide which we were able to show using stacked bar charts in place of the previous bar charts (Figures 2d, 3d, 4d, 5d). No extra figures have been added.

In addition to the revised document, we have uploaded a revised document with changes tracked. Line numbers of both documents are included in our responses, the numbers of the tracked changes document are in brackets. We have also included a tracked changes version of the revised Supplementary Information.

Reviewer #1 (Remarks to the Author):

This manuscript presents the analysis of food-related biodiversity footprints, it considers both land-driven footprints and GHG-driven footprints and two biodiversity metrics. The manuscript is very well-written, clearly structured and with high quality figures. However, I think it does not add sufficient novelty in its methods and findings. Therefore I cannot recommend this manuscript for publication in Nature Communications.

I hope the authors find my comments below useful.

General comments

My main issue with the manuscript in its current format is that it does not add sufficient novelty to the current state of the art. Marquardt et al. 2019 compared different biodiversity footprint metrics between them and also with land footprint. However the authors of this manuscript do not cite this work

(<https://www.sciencedirect.com/science/article/abs/pii/S1470160X19302687?via=ihub>). Wilting et al. 2017 quantified land-driven and GHG-driven biodiversity footprints. However their study is more complete since they cover more sectors in their GHG-driven footprints (<https://pubs.acs.org/doi/10.1021/acs.est.6b05296>).

Thank you for pointing out this oversight. We agree that we should have referred to Marquardt et al 2019. We have now remedied this both in the introduction (lines 74-76 (84-85)) and in the discussion (lines 596-599 (636-640)) as discussed later in our responses to the Specific Comments.

However, although our analysis builds on that by Marquardt et al and by Wilting et al, we respectfully disagree with reviewer 1 that our study is not sufficiently novel. The differences between species richness and rarity-weighted richness footprints are a key part of our analysis – this adds a valuable addition to the biodiversity footprint comparisons made by Marquardt et al 2019

since their paper does not analyse footprints based on rarity-weighted richness. Moreover, our analysis has greater spatial sensitivity than Wilting et al's since it has the novel combination of (i) allowing sensitivity to land use to vary by land-use type and biome; (ii) allowing for the natural variation of species richness across political regions; and (iii) allowing for spatial variation in future temperature change and in species' responses to that temperature change. Our in-depth focus on the production and consumption of food-related products complements Wilting et al's broader sector approach, addressing a different suite of questions. We use a different approach to Wilting et al and believe that it is important that land-driven and GHG-driven biodiversity impacts are further compared using different trade models and biodiversity metrics. Indeed, our discussion section covers these important points of comparison (section 4.1.). Furthermore, we have now split our GHG-driven footprint into the impacts driven by carbon dioxide, methane and nitrous oxide and show methane to contribute to 70% of the total food-related production footprint, thus providing further novelty to the manuscript.

We now emphasise this novelty more clearly in lines 77-92 (86-102).

I have some questions regarding the methods used that need to be clarified (see Specific comments).

Thank you for these comments, they were very constructive – we have now clarified the methods (see responses below in the Specific Comments).

In the policy implications section it would be interesting to have a more targeted and in depth discussion on the application of the work to the current policy frameworks. This would add novelty to the work, a lot of developments in policy arena have happened since the publication of other biodiversity footprint related work.

We agree that the policy implications of the work are interesting and we have now added to this section, discussing how the work could be applied to sustainability regulations in trade deals and highlighting that our results showing the high footprint due to methane emissions provides further support for policies regarding dietary shifts and the transition of farmers away from livestock (lines 619-714 (663-770)).

Specific comments:

Line 145 – 147 – not clear which impacts refer to food waste. Can you please clarify?

We have now made it clear that the 'food waste' impacts refer to the treatment of food waste (lines 160-162 (173-175)).

'...impacts of food waste refer to the waste treatment and decomposition processes, and do not include the production impacts of the wasted food.'

Line 147 – 149 – you mention that “we regard land use as a one-off cost that results in a change in biodiversity and we assume that once land has been converted, it can be used repeatedly without the biodiversity cost increasing”. I don't understand how can the impact be a one-off but then calculated for an yearly economic flow. Does this mean that if you would calculate the impacts for

2012, there will be almost no impacts since they were attributed to 2011? I don't understand this sentence, could you please clarify?

Ensuring the reader understands this is really key so thank you for bringing this up. EXIOBASE provides the land area used in each year. So in order to calculate the change in land use costs between 2011 and 2012 using EXIOBASE, you would subtract the 2011 land use from the 2012 land use. If you wanted the 2012 land use you would use the figures for 2012. What we were trying to explain is that the biodiversity impact of land use arises due to conversion from natural habitat. The land used in 2011 may have been converted one year ago or 1000 years ago and we are assuming that whenever that conversion occurred, the cost to biodiversity is the same. This is what we meant by the term 'one-off cost'. In contrast, emissions accrue year on year. Using EXIOBASE to compare biodiversity loss from land use to GHGs in 2011 means that you are comparing biodiversity loss that has occurred over centuries (land use) to biodiversity loss that will be caused by a single year of emissions. It's critical that the reader understands this hence why we provide the explanation in lines 147-149. However, we may perhaps have over-complicated our explanation. We have re-written it as follows (lines 162-171 (178-188)):

'We calculate the biodiversity change associated with all of the land area used in food production in 2011 and assume that the biodiversity change associated with land conversion is immediate. GHG emissions that are released during land conversion are not considered, because, without detailed land-history knowledge, we cannot estimate the proportion of emissions that have dissipated since conversion, nor apportion food-production emissions across years. To put this gap in our coverage of GHGs into context, direct emissions from agriculture contribute 5.1-6.1 Pg CO₂-eq./yr while the clearing of native land for agriculture contributes around 5.9 (SD 2.9) Pg CO₂-eq/yr³⁴. Consequently, our ratio of land-driven to GHG-driven biodiversity change compares the impacts of the centuries-long process of global agricultural land conversion to the impacts associated with just a single year of GHG emissions.'

Line 174 – nice figures, it would be more clear if you add the units in all squares so one can follow. It would make it easier to understand the final unit and how it is used.

Units have now been added to all squares (please note we were not able to track changes for this.)

Line 189 – 190 – Where all of these data points/studies used? I suppose you filtered the PREDICTS database to select only the metrics relevant to you, and a selection of studies with primary vegetation and another land use type. Please specify how many data points were used in the end. In the SI you should also identify this per model. It would be also nice to add a map with the geographical distribution of the studies used.

This information along with the map has been added to the Supplementary Information (pages 4-6).

Line 197 – do you follow an approach presented in any other biodiversity footprint paper? If yes

please refer to it in this section. If not, can you explain what are the novelties and differences in your approach?

Thank you – reading back we realise we did not make the novelty of our study sufficiently clear. We have now updated the introduction to emphasise the novelty (lines 77-92 (86-102)):

‘We build on these prior analyses, introducing three novel aspects. (i) We calculate the biodiversity impacts of agricultural land use and GHG-emission footprints using models that directly output metrics of terrestrial biodiversity change in the same units, allowing the drivers’ impacts to be compared and splitting emissions into carbon dioxide (CO₂), methane (CH₄) and nitrous dioxide (N₂O). (ii) We consider change in local rarity-weighted species richness relative to an unimpacted baseline in addition to local species richness. Species richness, although easy to measure, captures only one of the many dimensions of biodiversity, and does not always decline with global biodiversity loss³³. Rarity-weighted richness gives greater weight to species with small geographic range size (range size correlates with species extinction risks³⁴) and so declines if rare species are replaced by more common ones. (iii) We use biodiversity models that allow us for the first time to capture regional variation in the sensitivity of biodiversity both to land-use differences and to climate change³¹. We base our biodiversity metrics on local measures of biodiversity averaged across the relevant agricultural areas as opposed to a value averaged across an entire exporting region, meaning that we better account for the wide variation in species richness that occurs within regions. Nevertheless, there will still likely be substantial variation in biodiversity responses within our agricultural aggregations.’

Line 216 – not clear why and how you determine the amount of area needed to produce €1M of product.

We take the area directly from EXIOBASE and have now made this clear in the text (lines 233-234 (251-252)):

‘where $A_{i,k}$ is the area of agricultural land used to produce €1M of product in land-use type i in region k , as given by EXIOBASE’

€1M is the unit of production, as explained in lines 127-129 (141-143):

‘ f is the $(1 \times im)$ direct intensity vector, representing the environmental pressures (e.g. area of land, mass of CO₂ emissions) associated with one unit (€1M) of production for each product sector in each region.’

Line 221 – I don’t understand the units. Normally, the characterization factor represents the impact per unit of pressure (species lost/km² of land use). If the unit of the CF is species x km²

when you multiply it by km² (from EXIOBASE) a different unit is achieved. This needs to be clarified.

We made our biodiversity footprint in equivalent units to those in the satellite table so it is the biodiversity cost of producing 1 Million Euros of product and already incorporates the land area cost. We have now made this clearer in the text (lines 230-242 (248-260)):

“The characterisation factor $CF_{i,k}$ was then calculated as:

$$CF_{i,k} = \Delta S_{i,k} \times A_{i,k}$$

(Equation 4)

where $A_{i,k}$ is the area of agricultural land used to produce €1M of product in land-use type i in region k , as given by EXIOBASE).

The characterisation factors for species richness have units of number of species \times km² and can be thought of as the count of the species lost, with this loss extending over the area of land required to produce €1M of product, as given in EXIOBASE”

Line 228 – Why final demand? Shouldn't you compute the emissions associated with production activities so that it is compatible with the environmental extension.

You are absolutely right – by ‘final demand’ we meant consumption but actually the first step is indeed to calculate the production (which is what we did). We have corrected the text, changing ‘final demand’ to ‘production’ (line 247 (265)).

Line 238 – You choose to use Global Temperature Change Potential instead of the most commonly used approach using Global Warming Potential. Why is that the case? Please justify why you do not use GWP. I would also recommend calculating the impacts using GWP for two main reasons, first is that it is the most commonly used metric, which would increase the comparability of your results and second as a sensitivity analysis.

We have added an explanation for our choice of GTP (lines 257-264 (276-283)):

‘GHG-induced warming is often described in terms of the Global Warming Potential (GWP), a metric which compares the radiative forcing integrated over a time period caused by the emission of 1 kg of an agent relative to the integrated forcing caused by the emissions of 1 kg of CO₂⁵⁰. However, GWP has been criticised as it does not translate into a climate response that is intuitively understood⁵¹.

We therefore calculated the increase in global surface temperature due to GHG emissions using the Global Temperature Change Potential (GTP), a metric designed to be an intuitive measure of climate response⁵¹ and one that has been used in previous biodiversity footprint studies²⁹.’

We do not think that adding a comparison using GWP would add value, because GWP has a unit of W m⁻² yr kg⁻¹ and does not directly translate into a well-known climate response, unlike GTP with its

unit of deg kg⁻¹. Moreover, previous studies estimating biodiversity footprints associated with GHG emissions have also used GTP (e.g. Wilting et al 2017).

Line 241 – The choice of 20 years period should be better justified. If methane accounts for only a small percentage of the total gases I would argue that you should choose a different time horizon.

Methane's contribution to GTP is 80 times that of CO₂ over a 20 year horizon and, as we now show, it constitutes 70% of the GHG-driven footprint so it is important to account for its effects in short-term warming. A 100 year time horizon was originally chosen arbitrarily (Shine 1990) and does not marry well with the short-termism of most political decisions. We have added more justification for our choice of 20 years (lines 266-270 (285-289)):

'Studies often use a time horizon of 100 years following the Kyoto Protocol, but this choice was originally made on an arbitrary basis and is not the most appropriate for shorter term continental climate responses as it masks methane's potency^{11,52}. Instead we chose a 20-year time horizon, to capture warming due to the relatively short-lived methane emissions, and to represent a time that is tangible to today's policy and decision-makers.'

Line 251 – is the use of RCPs consistent with the use of GTP?

Yes, it is. GTP gives us the expected temperature increase associated with each unit of product. The projections of species distributions under the RCP scenarios allow an estimation of the biodiversity change associated with that unit of product by putting temperature change in context with wider climate variables. We have now clarified this in the text (lines 278-281 (297-300)):

'We estimated the sensitivity of biodiversity to climate change based on future projections of changes in the distributions of terrestrial vertebrates under the Representative Concentration Pathways (RCP) climate-change scenarios⁵, thus putting our projected temperature change in context with wider climate variables.'

Line 255 – In the SI file you mention that "Species' projected responses to climate change were derived from¹⁰". What was the exact relationship? Can you add this information in SI?

We have added information about the species distribution modelling algorithms we used to the SI. The section that we changed now reads as follows (page 2-3):

'Projected changes in the distributions of species' (and thus changes in local species richness across terrestrial areas) as a result of climate change were derived from¹⁰. The response of species' distributions to climate was captured using five different species distribution modelling algorithms (BIOCLIM, DOMAIN, Maxent, Random Forests and Generalised Linear Models). These models each related species' observed distributions according to the IUCN Red List¹¹ and Birdlife International¹², to four climatic variables shown in previous studies to be strong correlates of animal distributions: minimum temperature of the coldest month, total annual precipitation, growing degree days and water balance, derived from the Worldclim Version 1.4 database¹³, which captures average climatic conditions for the period 1961-1990. BIOCLIM fits relationships between distribution records and

climatic variables using a bounding-box approach in niche space, DOMAIN by comparing the climatic similarity between observed occurrence points and potentially inhabitable areas, random forests using a machine-learning approach to identify climatic patterns in species' occurrence records, while generalized linear models and Maxent use classical parametric statistics or a maximum-entropy approach, respectively, to fit linear and quadratic relationships between species' occurrences and the climatic variables.'

Line 265 – 266 – you mentioned before emissions from final demand, but here you mention emissions from food production of each product p in each region k. It is not clear what emission were used, please clarify.

Thanks, indeed we have now corrected this and changed 'final demand' to 'production'.

Line 269 – please provide more information on the relationship used.

We have added the following sentence (lines 299-301 (318-320)):

'This grid is based on projected changes in the distributions of species under climate change, and describes the expected change in local species richness in any terrestrial location associated with a temperature increase of 1°C.'

Line 272 – please clarify the unit (see comment referring to line 221)

See earlier comment: The unit is species * km² and there is no need to further multiply it by area. Lines 238-242 (256-260):

'The characterisation factors for species richness have units of number of species × km² and can be thought of as the count of the species lost, with this loss extending over the area of land required to produce €1M of product, as given in EXIOBASE'

Line 285 – I would say that considering land use as a one-off impact is not compatible with the analysis of the economic flows of one year.

We appreciate we have caused some confusion here. Let's take an example of paddy rice. Let's say an area of forest is converted to paddy rice in the year 2000. If rice is grown in this same area, year on year, the cost to biodiversity from land use is assumed not to increase after the initial conversion of that land to rice paddy. However, the cost from GHG emissions will increase year on year as emissions are produced annually. In our analysis, we are calculating the impacts of food production on biodiversity in 2011 so we are using the land-driven costs of all land that was used in agriculture in 2011 and the GHG costs of all emissions produced in 2011.

To clarify potential confusion due to our term 'one-off cost', we have amended the paragraph accordingly (lines 317-324 (336-344)):

'We calculate the biodiversity change associated with all land used in food production in 2011, regardless of the year of conversion. Our measure of GHG-driven biodiversity change is associated

with emissions produced in 2011. We consider the impact of land use to be reversible and view land conversion as a 'one-off' cost, i.e. once land is converted biodiversity change is immediate and does not increase through time. In contrast we view GHG emissions as irreversible, repeated annual costs that occur 20 years after emission. We would expect the global biodiversity loss caused by a single year of emissions to be much lower than that caused by the conversion of the total amount of agricultural land used in 2011.'

Line 343 – Does this mean that the countries in the top 10 positions are the same for all the different metrics analyzed?

No - some regions are in the top 10 for one footprint but not another. To clarify, for each relevant figure legend, we have changed it to "Regions which are in the highest ten for one or more footprints are shown."

Line 370 – 372 – Why does a ratio of 100 indicates that reducing GHG biodiversity footprint is a priority?

Following the correction of our code, the ratios have lowered. We have re-written and clarified this sentence accordingly (lines 362-369 (386-393)):

'The ratio of land-driven to GHG-driven biodiversity loss varied by region from 16 for rarity-weighted richness production footprint in Russia to 855 for production in RoW C&S America, with several regions, including China, India and RoW Asia, having ratios around 50. Finding ratios of 50 or lowers is concerning as it shows that direct emissions from a single year of a region's food production will cause biodiversity loss equivalent to 2% or more of the biodiversity loss caused by that region's total historic land use. Furthermore, we substantially underestimate biodiversity losses from GHG emissions since our analysis does not include emissions from land clearance.'

Line 390 – I wonder if this is a real result of somewhat related with uncertainties of the MRIO table. Please check this papers to better understand if this is the case for this result:

<https://onlinelibrary.wiley.com/doi/10.1111/jiec.12833>

<https://journalofeconomicstructures.springeropen.com/articles/10.1186/s40008-020-0182-y#Sec20>

We thank the reviewer for bringing our attention to the Giljum et al paper, with which we were not familiar. The discrepancy between the estimates stemming from different MRIOs for Taiwan's material flow is very interesting. EXIOBASE and ICIO's estimates are over double that of Eora's. Obviously at least one of the MRIOs is giving an inaccurate estimate but it's impossible to say which one.

The footprints calculated in the Giljum et al paper are consumption-based footprints per capita whereas the footprint we are commenting on is a production-based footprint per area. However, as

we show in Figure 2, regions' production and consumption-based footprints tend to be similar so it seems fair to assume that if Taiwan's consumption-based footprint has been over-estimated then so has its production-based footprint.

We have added the following bracketed phrase to the end of the results paragraph (lines 448 (482-483))

'(although see Giljum et al 2019 regarding possible over-estimation of Taiwan's material footprint).'

We now also similarly refer the reader to Giljum et al 2019 in the Discussion (lines 680-681 (735-736))

Line 418 -420 – I would expect Brazil to have a high per capita rarity-weighted footprint. Why is that not the case? Can you please explain further?

Brazil has the 6th highest per capita RWR footprint so it's not low, we are just pointing out that it is lower than RoW C & S America's. Brazil is consuming products which on average have a higher species richness footprint than RWR footprint. If you look at Supplementary Figure 8 you can see that Brazil's per capita consumption-based species richness footprint is much higher than RoW C&S America's for cattle, processed cattle, dairy, vegetable oils and processed sugar.

We have added the following bracketed sentence to explain this point (lines 474-475 (509-511):

'(Brazil has particularly high per-capita species richness footprints relative to RoW CS America for cattle, processed cattle, vegetable oils and dairy, see Supplementary Figure 8).'

Line 470 – 471 – If I am not mistaken in all figures for all the footprint the top 10 or 5 countries with highest footprints are the same, irrespective of the type of footprint. If this is true, doesn't it make it less important which footprint to use?

The regions with the highest footprints are not the same for each metric, hence why there are more than 10 regions in the figures (i.e., in all panels, we show all metrics that were in the top 10 for any of the metrics). We clarified the captions to make this clearer:

"Regions which are in the highest ten for one or more footprints are shown."

Line 497 – You don't mention the following studies:

<https://www.sciencedirect.com/science/article/abs/pii/S1470160X19302687>

<https://www.sciencedirect.com/science/article/abs/pii/S1470160X19302687>

<https://conbio.onlinelibrary.wiley.com/doi/full/10.1111/con4.12321>

Thank you for pointing this out – we were familiar with the work of Marquardt et al 2019 and it was an oversight in not citing this work. We cannot compare our results to Marquardt et al's since their study covers all products and not just food. However, we do now cite their work (line 602 (643)) when explaining that our discrepancies with Chaudhary and Kastner are likely in part due to differences in biodiversity metrics.

Similarly, much of Kitzes et al's study is based on total goods, not just food related products. However, they do one analysis that breaks goods down by product so we have included this in our comparison section (lines 592-594 (632-634))

'Kitzes et al. (2017), using metrics based on birds and a greater spatial disaggregation than our study, also find particularly high impacts of bovine products and, in contrast to our results, of processed rice.'

Line 512 – 518 – These conclusions are misleading since Chaudhary and Kastner did not use an MRIO model which might have more influence on the results than the biodiversity metric used. Please revise this.

Thank you for pointing this out. We had explained that differences between analyses are likely to result, at least in part, from different trade models (lines 598-601 (638-642)) but we have now explained that Chaudhary and Kastner used bilateral trade data (lines 579-580 (617-618)).

'Chaudhary and Kastner¹⁶ use bilateral trade data in combination with a metric of the number of species committed to extinction.'

Line 519 - %21 – I don't think you can state this without a proper analysis. Also check the literature on the differences of the results obtained with physical and monetary approaches. For example: <https://www.sciencedirect.com/science/article/abs/pii/S0921800913003583> <https://www.sciencedirect.com/science/article/abs/pii/S0921800915000932>

We have now cited Marquardt et al 2019 who show that using the PREDICTS data (which we use, in our case, making the important addition of allowing for biome and region sensitivity) versus using the PDF metric, which Chaudhary and Kastner use, does lead to different results (lines 74-76 (84-85)). We therefore feel it is fair to suggest that the different biodiversity metrics that we use are likely to account for some of the differences in results. However, we are very aware of how easy it is for differences to arise solely due to trade models and do also mention that trade models will also account for some of the differences in the results. We now cite Kastner et al 2014 and Bruckner et al 2015 (lines 598-599 (638-639)). We have revised the sentence as follows and moved it to the end of the paragraph so that it refers to the differences between all the studies we mention: lines 579-580 (617-618)).

'The discrepancies between studies will in part result from differences in trade models^{61,62} but are also likely to result from differences in the biodiversity metrics used²⁸, adding support for ours and Marquardt et al.'s²⁸ findings that different biodiversity metrics lead to different conclusions.'

Line 530 – 536 – See my previous comments on the methods. I suggest you follow the same approach (at least as sensitivity) so that the comparison is possible.

Thank you. As we explain above, we do not think that adding a comparison using GWP would add value. Previous studies (e.g., Wilting et al., 2019) have also used GTP, as we do. More fundamentally, GWP has a unit of $\text{W m}^{-2} \text{yr kg}^{-1}$ and does not directly translate into a well-known climate response unlike GTP with its unit of deg kg^{-1} . Wilting et al 2017 use integrated GTP (which is essentially calculating the warming due to sustained emissions over 100 years as opposed to a single pulse of emissions from one year.)

We don't feel that integrated GTP (as used by Wilting) is the appropriate measure to use in our analysis (we are interested in the impact of a single pulse of emissions, not sustained emissions over a period of time) but were we to use it, as we explain in the discussion (lines 608-609 (651-652)), we would expect the values to be higher than ours. It has hard to estimate exactly how much higher – integrated GTP over a 20 year horizon would be approximately 100 times our measure since it is simply GTP (which we use) integrated over a period of 100 years. Wilting et al used a time horizon of 100 years meaning their measure will not have captured all of the warming from methane and so we would expect their values to be less than 100 times our measure, which indeed they are.

Line 566 – I think you cannot really make this statement since you focus only on food products.

This was careless wording on our part – we have changed the statement to 'different decisions with respect to the sustainable trade of food products' (line 635 (679)).

Reviewer #2 (Remarks to the Author):

I reviewed the analysis title „Impacts of global food supply on biodiversity via land use and climate change”.

The submitted manuscript investigates new methods by combining different datasets and using an EEMRIO to calculate land- and GHG-driven biodiversity footprints globally. Overall, the analysis is interesting, novel, and well-structured, however, there are a couple of issues and open questions that need to be addressed.

Line 46: But how climate change affects biodiversity globally is regionally still different, right?! In some regions with average colder temperatures, like Canada or Siberia, there might be more biodiversity evolving due to climate change?!

Yes, absolutely. This is a really good point – in some locations species richness is actually increasing due to climate change. In terms of the structure of the paper we think it's best to keep things general at this early stage in the introduction but we draw attention to the regional differences in climate change further on (lines 283-285 (302-304)). With regard to line 46 we have changed 'biodiversity loss' to 'biodiversity' since as you point out, it is not always a loss. We have also changed 'biodiversity loss' to 'biodiversity change' (line 243 (261)) and made it clear we allow for non-uniformity of warming across the globe (lines 283-285 (302-304)).

Line 49: Either here or later at line 89 you should differentiate and describe production vs consumption footprints.

Thank you for suggesting this. At line 89 (now lines 95-98 (105-108)) we have added

‘Production-based footprints are based on the total impacts associated with the products produced within a region, whereas consumption-based footprints are the total impacts associated with the products consumed within that region.’

Lines 86-88: For clarification, it would be helpful for the reader if you could state that your data are building on spatially explicit information but is aggregated and averaged to reflect heterogeneity but, in the end, do not represent spatially explicit information any more.

We have changed the sentence accordingly lines 88-92 (98-102):

‘We base our biodiversity metrics on local measures of biodiversity averaged across the relevant agricultural areas as opposed to a value averaged across an entire exporting region, meaning that we better account for the wide variation in species richness that occurs within regions. Nevertheless, there will still likely be substantial variation in biodiversity responses within our agricultural aggregations.’

Lines 95-97: Do you refer to both production and consumption footprints here

In the majority of cases, yes, we are referring to both production and consumption footprints here. We have now made it clear where we refer to production or consumption footprints only (lines 103-105 (113-115)).

‘We also explore biodiversity footprints per km² (production) and per capita (consumption) for each region and look at the proportion of regions’ consumption footprints that are imported.’

Line 117: When using Y from EXIBOASE the matrix is 7 x i, right? Because there are 7 different final demand categories. However, not sure if you need to mention that when you aggregate into one final demand category for each region. You could say “associated with the final demand of households, governments, capital formation, ... of each region.”^a

Thank you – this is a really good suggestion. We have amended the sentence to (lines 134-136 (146-148))

‘Y is the ($im \times i$) matrix of final demand (associated with households, non-profit organisations, governments, capital formation, changes in inventories and valuables and exports) given in monetary terms (€1M).’

Line 130: “spare vector populated in the entries for ...” -- Sounds a bit odd. It shows which sectors/production activities require direct land-use. The same allocation principle applies to GHGs, right?

There is a difference between the vectors for land use and GHGs. GHGs are associated with production in all sectors but there are only values of land use impacts for sectors which directly involve cropland or pasture e.g. values for paddy rice but not for processed rice. Hence the use of

the term 'sparse' referring to a vector that has a relatively small number of non-zero elements. (This terminology is used in other papers describing MRIOs, e.g. Marques et al 2019.)

We have added a clarification in the text (lines 139-141 (151-153)):

'The vector relating to land use is a sparse vector populated in the entries for production activities that directly involve use of cropland or pasture, e.g. populated for paddy rice but not for processed rice.'

Lines 136-137: I understand, but I still have to flag this, it is quite old data and I'm unsure if 12-year-old data represent a proper fit to present data, also since you derive policy implications and not only methodological proof of concept. There are no land-use or spatial crop data available after 2011?

We agree that using data from 2011 is far from ideal. Unfortunately the most recent global crop data are from 2010 (MapSPAM) (and some from as far back as 2000 (EarthStat)). Whilst EXIOBASE has more recent trade data, EXIOBASE land use data go no further than 2011. It's a problem which is regularly discussed and lamented amongst colleagues!

We do explain the crop data used further down in the text and discuss these land use limitations in the discussion but we have now added in citations at this point to make it clearer for the reader (lines 146-148 (158-161)):

'We chose 2011 as our year for analysis since land use data are available up to 2011 only (<https://zenodo.org/record/5589597#.YnkHvOjMK3A>) and spatial data for the majority of crops^{36,37} does not extend beyond this date (see below).'

Lines 147-149: How do you do that? A share of the upstream land-use flows based on spatial explicit information of land-use?

Reviewer 1 was also confused by this. We have changed the sentence accordingly (lines 162-164 (178-180)):

'We calculate the biodiversity change associated with all of the land area used in food production in 2011 and assume that the biodiversity change associated with land conversion is immediate.'

Figure 1a): So, for each biome you calculated a mean species richness/rarity richness and multiplied by the total land-use area reported in EXIOBASE?

Yes, that is correct.

Lines 177-178: You are only including animals, no plants. Would be good to mention somewhere.

Yes, this is a good point, thank you. Our land use models represent plants but the climate change model does not. We have added the following sentence to the Limitations section in the Discussion to make this clear (lines 732-735 (788-791)):

‘Our models of land-use impacts represent terrestrial vertebrates, invertebrates, plants and fungi but, due to data limitations, our biodiversity metrics used to estimate climate impacts refer to terrestrial vertebrates only.’

Lines 227-229: How did you calculate “GHG emissions associated with each region’s 2011 final demand for the 33 different food-related products.”? Which method?

Reviewer 1 pointed this out as well – we worded this misleadingly and have now changed ‘final demand’ to ‘production’ and explained that we used EXIOBASE (lines 247-248 (265-266)).

Lines 232-236: Again, what is the method for this – the formular? How did you calculate the GHG footprints for 33 different products? Output multiplier?

Yes. We have added “Footprints were calculated using the environmentally-extended Leontief model described in section 2.1.” (lines 255-256 (274-275))

Lines 232-233: You don’t mention the expression of CH4 and N2O in CO2 equivalents.

We give the GTP values for each gas in the following paragraph (lines 27-272 (290-291)).

Lines 235-236: Also the N2O from combustion and agriculture?

Yes – and the CO2! Thank you for pointing this out. We have now changed the sentence to ‘Emissions were summed over the four categories given by EXIOBASE (‘Combustion – air’, ‘non-combustion’, ‘agriculture’ and ‘waste’) (lines 254-255 (274)).

Lines 240: And at a given space/biome etc?

We are not entirely certain what you mean here. If you are referring to the place of emission then no, the GTP will be the same wherever the emission occurs. If you mean that warming is not uniform across the globe that is of course correct. The GTP does not take this into account – it’s an average global measure. We weight the GTP at a later stage using projected local temperature changes (described further on in the methods) to allow for the non-uniformity of warming across the globe (lines 285-301 (304-320)).

Lines 241: Given that you use data from the year 2011 (provided that they appear not too old for publication) you could check how the GTP model captured the actual changes in global temperature change; since 12 years of your 20-year time horizon have already passed.

It did occur to me that we’re actually pretty close to the end point of the change we are predicting. It would be difficult to ground-truth though since we are only calculating the warming from a single year of emissions from food products which works out on the order of 10^{-3} degrees of warming!

Lines 285-286: Except for land use change that occurred in 2011 or the year before?

But how does this consideration affect your results and how you allocate emissions from land use to products?

Reviewer 1 also was confused here. We have re-written the section, hopefully more clearly (lines 316-324 (335-344)):

‘However, there are differences in the way that the two drivers impact biodiversity, and in the methods we used to model these impacts. We calculate the biodiversity change associated with all land used in food production in 2011, regardless of the year of conversion. Our measure of GHG-driven biodiversity change is associated with emissions produced in 2011. We consider the impact of land use to be reversible and view land conversion as a ‘one-off’ cost, i.e., once land is converted biodiversity change is immediate and does not increase through time. In contrast, we view GHG emissions as irreversible, repeated annual costs that occur 20 years after emission. We would expect the global biodiversity loss caused by a single year of emissions to be much lower than that caused by the conversion of the total amount of agricultural land used in 2011.’

Lines 306-307: You didn’t yet explain what the difference is and what they show (or I missed it).

We now define the two types of footprints in the introduction (lines 95-98 (105-108))

‘Production-based footprints are based on the total impacts associated with the products produced within a region whereas consumption-based footprints are the total impacts associated with the products consumed within that region.’

Lines 321: Could you provide the total values for the GHG footprints of all food-related products, globally and/or nationally? This number could be compared to the numbers mentioned in the introduction in lines 31-32.

We have added the following paragraph to the results section (lines 377-383 (402-408)):

‘Methane emissions account for 70% of the total GHG-driven biodiversity footprint from food-related products, compared to 42% of the GHG-driven footprint of all EXIOBASE’s products. Carbon dioxide contributed to 18% of food’s total footprint versus 54% of the footprints of all products and nitrous oxide contributed to 12% of food’s footprint and 4% of the footprints of all products. Food-related products account for 23% of the total emissions in EXIOBASE and the total GHG-driven footprint of all food-related products was approximately 1% of the land-driven richness footprints.’

Lines 333-334: So, with their food system production they harm their domestic biodiversity as well as global biodiversity (GHG-driven). This could be mentioned to illustrate and differentiate.

We have changed the sentence to: (lines 371-374 (395-398))

‘RoW Asia & Pacific, RoW CS America, Brazil, Mexico and South Africa are all net exporters of both land-driven and GHG-driven biodiversity meaning that international trade is harming both their domestic biodiversity and, via climate change, global biodiversity.’

Lines 342-343: I still don’t know the difference between production and consumption footprints, but they appear remarkably close to one another. That is not production- vs consumption-based approaches?!

We now define production vs consumption footprints in the introduction (lines 95-98 (105-108)). Yes, it is the same as production-based and consumption-based accounting. The production and consumption footprints are similar and initially we were surprised by this too. Other studies show similar results, e.g. Dalin et al., 2017 (<https://www.nature.com/articles/nature21403>), for water-scarcity footprints. It is important to remember that: (1) that the footprints shown in Figure 2 are mostly associated with very large regions, and so domestic consumption will be high; (2) a lot of trade occurs within the Rest of World regions depicted in the figure, for example African countries trading with each other – because these regions are aggregated this trade is not picked up in the figure; and (3) the footprints are summed across all food products and so may balance each other out – if we look at individual products, as shown in Supplementary Figure 5, we see that production and consumption footprints can show more variation than in Figure 2. For example, the production footprint of oil seeds in RoW C&S America (WL) is more than twice the consumption-based footprint (shown below). The opposite pattern occurs in China.

Also, I realize these are total values, but I would recommend to at least mention why there are no European countries shown nor discussed.

We have added the following sentence (lines 375-376 (400-401)):

‘The top ten footprints stem from regions with very large land areas and/or populations and, aside from Russia, do not include regions in continental Europe.’

Lines 359-361: I don’t see this in e) or f). Isn’t it Africa and C&S America that stand out? Or where can I find that information?

Thank you for highlighting this – we made a typo here and should have written N America instead of E Europe. ‘Stand out’ was also a poor choice of words. We have removed this sentence since it was not obvious what we were trying to say and we discuss the land-driven:GHG driven footprint ratio in a subsequent paragraph.

Lines 440-442: If correct, you could state here, again for illustration, that these countries are relatively biodiversity poor but highly industrialized in their land use system.

We have added this observation (lines 502-507 (538-543))

‘We found a near-reversal of net imports (i.e. imports minus exports, or consumption-based footprint minus production-based footprint) for land-driven versus GHG-driven biodiversity footprints (Figure 6, Supplementary Figure 9). The United States, the United Kingdom, Germany, Russia, Japan, China and RoW Middle East – regions that tend to be relatively biodiversity poor and highly industrialised in their land use systems – are all net importers of land-driven biodiversity loss but net exporters of GHG-driven biodiversity loss.’

It would be useful to also mention how net-trade is calculated and what negative and positive values represent.

Yes, good point, thank you. We have now added this in brackets (lines 502-504 (538-540)):

‘We found a near-reversal of net imports (i.e. imports minus exports, or consumption-based footprint minus production-based footprint) for land-driven versus GHG-driven biodiversity footprints (Figure 6, Supplementary Figure 9).’

Lines 485-486: However, land-use intensity is still not reflected (probably early on in the manuscript). You should at least mention this and that other indicators, like HANPP (and embodied HANPP) can account for that.

We now mention that we do not consider agricultural intensification in the methods (lines 154-158 (167-171)):

‘We account for land use intensity in the sense that the amount of land required to produce one unit of food differs between the EXIOBASE regions but, in common with previous biodiversity footprint studies, we do not model the effects of different agricultural practices (e.g. fertiliser and pesticide use) or different yields per unit area on biodiversity.’

We have amended a sentence in the discussion (lines 724-727 (780-783)) and added another sentence (lines 730-732 (786-788)):

‘Agricultural intensity is not yet captured within MRIO models of trade flows. Consequently, in common with other footprinting studies, e.g.^{26,27}, our analysis only measures biodiversity impacts caused by direct land-use change, and does not explicitly consider the impacts of agricultural intensification.’

‘New approaches, such as the Human Appropriation of Net Primary Production (HANPP), e.g.²¹, show promise for capturing effects of agricultural intensity in the future, although linking such measures to biodiversity loss remains a challenge.’

Lines: 502-503: Actually, it would be interesting to also see the aggregated effects of both in a figure and to see it discussed in the text.

Our original plan was to show the aggregated effects of land use and GHGs on biodiversity but then we realised that since the effects of GHGs are around 50 – 1000 times smaller than those of land use, the graph would look essentially identical to the land-driven footprint.

Line 514: You should not that there is an EXIOBASE version with higher national detail (n=214) but only captures land use as an environmental extension so far, not GHGs (see <https://zenodo.org/record/2654460>).

We have added the following sentence in brackets (lines 582-583 (620-621)):

‘An extension of EXIOBASE disaggregates trade regions into 214 countries, but its environmental extension has fewer land use categories and does not cover GHG emissions⁵⁸.’

Lines 550-551: And method, or resolution respectively.

In this particular paragraph we are discussing the biodiversity metric so we have now made this clear (lines 620-621 (664-665)).

‘Our study shows that measurements of trade-related impacts of biodiversity differ considerably depending on the biodiversity metric used.’

Lines 625-626: I’m not sure yet how you calculated the GHG footprints of food-related sectors/products. But since it’s a supply-chain perspective, also machinery that is required by food producing sectors would fall into that, right?

Emissions from machinery would indeed be included but it’s the fertiliser that makes the biggest contribution to the emissions.

We have amended the line to explain that emissions do also arise from combustion and have added the relevant citation (lines 707-708 (763-765)):

‘The emissions associated with crop production largely arise from fertiliser use (although also fuel combustion, industry and waste)³⁵.’

Lines 626-628: It would be interesting to see (from a comparison) if it was useful from a biodiversity footprint perspective to onshore production to N America and W Europe from other regions to reduce land-driven biodiversity footprints and increase GHG-driven biodiv footprints but still decrease the total biodiv footprint by regionalizing production for consumption.

That would be interesting. There are some studies that investigate source shifting and/or the optimisation of production areas. Being able to compare land-driven to GHG-driven footprints would be useful in this respect.

We have added the following sentence (lines 712-714(768-770)):

'Future studies could use our comparable land-driven and GHG-driven biodiversity metrics to investigate the reduction of biodiversity footprints via source-shifting and the spatial optimisation of cropland.'

REVIEWER COMMENTS

Reviewer #1 (Remarks to the Author):

General comments:

I feel the authors answered the previous comments appropriately. Thank you for that.

I still feel that the novelty of the work is not sufficient. I appreciate the splitting of the GHG footprints by different gases and I think it adds novelty, I would recommend exploring it more in depth, think perhaps of a dedicated results section.

I still have some comments that I would like to see addressed.

The authors should pay attention in their results to the Rest of the World regions and nec sectors. Normally these are regions and sectors with less quality data. See for example: <https://journalofeconomicstructures.springeropen.com/articles/10.1186/s40008-020-0182-y>

In the Results section the authors sometimes refer to RoW and sometimes not. Are these different aggregations? If yes please explain and make it explicit, if not please make it consistent.

It should be more clear in the results section the year of analysis. It should be mentioned in the captions of the figures.

In the Discussion I think it should be more clear what is meant by “appropriate metrics of biodiversity”. It should be more clear that the metrics analysed provide different insights into different aspects of biodiversity.

Specific comments:

Abstract – I would recommend be explicit about the year of analysis in the abstract

Line 68 – when you refer to the difficulty in allocating extinctions to different actors across time do you refer to past or future? Please clarify.

Line 72 – specify the version of the GLOBIO model used by Wilting et al.

Line 93 – please make explicit the year that your analysis refers to.

line 99 – 101 – This sentence is not clear. Please clarify.

Line 206 – 208 – This sentence can be misleading. Were all these data points used for the calculation of the characterization factors? Please make explicit here how many were used,

and not just in SI. Adjust Supplementary Figure 1 accordingly (if necessary).

Line 386 – Explain what RoW means in the legend of the figure. The difference in colours in panel d is not very clear. Perhaps add different colours to the different gases (this applied to all figures).

Line 397 – Not clear if these are RoW or aggregated world regions. Please clarify.

Line 399 – Is Asia & Pacific the same as RoW Asia and Pacific, please clarify and if needed homogenize.

Line 565 – You should have also account for different years analysed in different studies.

Line 744 – please specify which new methods you presented. As it is stated might be an over claim.

Reviewer #2 (Remarks to the Author):

I have reviewed the resubmitted version of this article and found that the authors have satisfactorily addressed all my concerns and questions. I therefore recommend the paper for publication without any further questions on my part.

COMMENTS

Reviewer #1 (Remarks to the Author):

General comments:

I feel the authors answered the previous comments appropriately. Thank you for that. I still feel that the novelty of the work is not sufficient. I appreciate the splitting of the GHG footprints by different gases and I think it adds novelty, I would recommend exploring it more in depth, think perhaps of a dedicated results section.

We have followed your recommendation and made a dedicated results section to discuss the different contributions of the GHGs (section 3.4, lines 499-517). We have moved the relevant results from the sections they were previously in and have also added description of some of the results contained within the supplementary information, explaining that emissions from animal-products, paddy rice and food waste are primarily driven by methane and those from ‘fish and other fishing products’ and fertilisers by nitrous oxide. We also show how, within one product, contributions can differ by region using wheat and fruit and vegetables as examples.

I still have some comments that I would like to see addressed.

The authors should pay attention in their results to the Rest of the World regions and nec sectors. Normally these are regions and sectors with less quality data. See for example: <https://journalofeconomicstructures.springeropen.com/articles/10.1186/s40008-020-0182-y>

Thank you for bringing our attention to this. Aggregating regions does of course underestimate the volume of trade – and thus the impacts embodied within trade - and both regional and sectoral aggregation can influence MRIO outputs, which we originally referred to briefly in the Discussion. We have now added to this to explain more fully, citing an example from the Bjelle et al 2020 paper you suggest (lines xx):

‘Sectoral and regional aggregation are particularly pertinent given that some of the highest footprints we find relate to the highly aggregated RoW regions and ‘Other food’ sector. Regional aggregation underestimates the volume of trade (and thus embodied impacts) and has been shown to under/over estimate the land use impacts of agricultural commodities by up to 20% and 10% respectively, compared to a disaggregated version of EXIOBASE with full country resolution⁵⁸.’

In the Results section the authors sometimes refer to RoW and sometimes not. Are these different aggregations? If yes please explain and make it explicit, if not please make it consistent.

In Section 3.2 of the results we aggregate EXIOBASE regions into world regions, hence why we do not refer to RoW regions here. Thank you for pointing out the confusion. We have now amended the section to make it clear for the reader:

Lines 394-395. 'Production-based footprints vary considerably among food-related groups and, within food-related groups, among aggregated world regions (Figure 3).'

We have also amended the relevant figure caption:

'Production-based footprints of aggregated food-related groups within aggregated world regions for the year 2011.'

We also aggregate world regions in Supplementary Figures 11 and 12 and have amended the text in their figure captions and in the relevant results paragraph (lines 557-559):

'We find that 10%, 15% and 8% respectively of land-driven species richness, land-driven rarity-weighted richness and GHG-driven richness footprints are embedded within trade between aggregated world regions (Supplementary Figure 11),'

It should be more clear in the results section the year of analysis. It should be mentioned in the captions of the figures.

We have added the year to all of the figure captions in both the main manuscript and the supplementary information. We have also added the year of analysis to the Results paragraph in which we first talk about GHG-driven footprints (line 366). (We already mentioned it in the opening land-driven footprint paragraphs of the results.)

In the Discussion I think it should be more clear what is meant by "appropriate metrics of biodiversity". It should be more clear that the metrics analysed provide different insights into different aspects of biodiversity.

Thank you for pointing this out. We have rewritten the sentence to make our meaning clearer (lines 687-689):

'It is important that such tools use metrics that capture different aspects of biodiversity since, if based on land use alone, impacts on small-ranged species in the tropics will be severely underestimated, particularly in CS America.'

Specific comments:

Abstract – I would recommend be explicit about the year of analysis in the abstract

This is a good point, we have now included the year in the abstract (lines 6-7).

'We use the multi-regional input-output model EXIOBASE to estimate the biodiversity impacts embedded within the global food system in 2011.'

Line 68 – when you refer to the difficulty in allocating extinctions to different actors across time do you refer to past or future? Please clarify.

We mean past and future and have clarified accordingly, also changing 'actors' to 'drivers' (lines 68-69):

‘it is unclear how these extinctions would be allocated to different drivers across time, both past and future’

Line 72 – specify the version of the GLOBIO model used by Wilting et al.

We have added that it was version 3.5 (line 73):

‘The GLOBIO 3.5 biodiversity model³⁰, on which Wilting et al.’s²⁹ analysis was based, assumes that effects of land use and climate change are even across all terrestrial regions;’

Line 93 – please make explicit the year that your analysis refers to.

Thank you, we have now included the year (line 95)

“We examine the international production-based and consumption-based footprints of food-related commodities in 2011 in terms of: a) land area; b) species richness (land-driven and GHG-driven); and c) rarity-weighted species richness (land-driven and GHG-driven).”

line 99 – 101 – This sentence is not clear. Please clarify.

Thank you, we agree this sentence was not clear enough. We have now rewritten it (lines 101-103):

‘We ask whether the estimated impacts we calculate using the different footprint types (i.e. land-driven versus GHG-driven; and land area versus species richness versus rarity-weighted richness) would lead to the same broad policy recommendations with respect to sustainable production, consumption and trade.’

Line 206 – 208 – This sentence can be misleading. Were all these data points used for the calculation of the characterization factors? Please make explicit here how many were used, and not just in SI. Adjust Supplementary Figure 1 accordingly a(if necessary).

Supplementary Figure 1 is correct and only contained the data points used for calculation of the characterisation factors. We have now given the exact number of locations and studies the characterisation factors were based on in the main text:

Lines 213-215: ‘allowing the response of biodiversity (i.e. species richness (based on 21,986 locations from 637 studies) and rarity-weighted richness (based on 15,198 locations from 403 studies)) to these land-use categories to vary among the six broad biome groupings listed above.’

Line 386 – Explain what RoW means in the legend of the figure. The difference in colours in panel d is not very clear. Perhaps add different colours to the different gases (this applied to all figures).

We have added ‘RoW = Rest of World’ to all the relevant figure captions.

At the time, we gave a lot of consideration as to how best present the contributions of the different GHGs and, in response to your comment, have reconsidered. In the case of Figure 2,

the figure is already using colour to differentiate between production and consumption footprints and we felt it was important that the colour scheme in panel d matched the colour schemes in the other panels in order not to cause confusion for the reader. For this reason, we used different shades of the same colour to represent the different GHGs. However, what was clear for us obviously was not clear for you, so we consulted with a couple of colleagues to see which aspect was unclear. They agreed with you in the case of Figure 2 and it seems that the lightest shade was the problem – we have therefore made it less opaque so that the difference between the lightest shades of red and blue is more obvious.

Similarly for Figure 3, (particularly since it already has so much colour!) we feel it would be best to keep a consistent colour scheme across panels. We have remade the figure making the lightest shade less opaque so the difference between the lightest shades is more obvious.

We would like to stick with the shading we have for the greyscale panels since our colleagues said that in these cases the shading scale worked well for them.

Line 397 – Not clear if these are RoW or aggregated world regions. Please clarify.

We have now clarified that these are aggregated world regions (see response to comment above).

Lines 394-395. ‘Production-based footprints vary considerably among food-related groups and, within food-related groups, among aggregated world regions (Figure 3).’

Line 399 – Is Asia & Pacific the same as RoW Asia and Pacific, please clarify and if needed homogenize.

Asia & Pacific is an aggregated world region and not the same as RoW Asia and Pacific. This should now be clear to the reader due to our amendment in lines 394-395:

‘Production-based footprints vary considerably among food-related groups and, within food-related groups, among aggregated world regions (Figure 3).’

Line 565 – You should have also account for different years analysed in different studies.

Thank you for pointing this out. Even if data from the same year are used, using different resolutions of region or sector can affect the results of MRIOs quite considerably which is why we mentioned this in line 565. However, of course using a different year would also affect the results. We have therefore added this in brackets (lines 589-591):

‘Comparing the results of our study with those of previous analyses of the embedded biodiversity footprints of food is not straightforward since studies generally differ in resolution with respect to regions and/or products (and may use data from different years).’

Line 744 – please specify which new methods you presented. As it is stated might be an

over claim.

We have amended the sentence accordingly:

Lines 768-771:

‘We have presented new methods that capture regional variation in the sensitivity of biodiversity both to land-use and climate change to estimate the land-driven and GHG-driven biodiversity impacts embedded within the production, consumption and trade of food-related commodities.’

Reviewer #2 (Remarks to the Author):

I have reviewed the resubmitted version of this article and found that the authors have satisfactorily addressed all my concerns and questions. I therefore recommend the paper for publication without any further questions on my part.

Many thanks for your original comments and for reviewing our revised manuscript.